# Construction of double reaction zones for long-life quasi-solid aluminum-ion batteries by realizing maximum electron transfer

Zhijing Yu[1,2], Wei Wang ®[1,2] ✉, Yong Zhu[2], Wei-Li Song[3], Zheng Huang[1], Zhe Wang ®[1] ✉ & Shuqiang Jiao[1] ✉

Achieving high energy density and long cycling life simultaneously remains the most critical challenge for aluminum-ion batteries (AIBs), especially for high-capacity conversion-type positive electrodes suffering from shuttle effect in strongly acidic electrolytes. Herein, we develop a layered quasi-solid AIBs system with double reaction zones (DRZs, Zone 1 and Zone 2) to address such issues. Zone 1 is designed to accelerate reaction kinetics by improving wetting ability of quasi-solid electrolyte to active materials. A composite three-dimensional conductive framework (Zone 2) interwoven by gel network for ion conduction and carbon nanotube network as electronic conductor, can fix the active materials dissolved from Zone 1 to allow for continuing electrochemical reactions. Therefore, a maximum electron transfer is realized for the conversion-type mateials in DRZs, and an ultrahigh capacity (400 mAh g$^{-1}$) and an ultralong cycling life (4000 cycles) are achieved. Such strategy provides a new perspective for constructing high-energy-density and long-life AIBs.

The exploration and utilization of renewable energy have been paid more and more attention due to the excessive consumption of fossil-fuel resources and serious environmental pollution[1]. The corresponding energy storage techniques, especially electrochemical batteries, are playing important roles to compensate for the intermittency in renewable energy production[2–4]. Rechargeable aluminum-ion batteries (AIBs) have been proposed as one of the most promising candidates for next-generation large-scale energy storage systems because of the abundant resources of raw materials and high safety[5,6]. In 1970s, the reversible aluminum deposition/dissolution processes were demonstrated in the non-aqueous chloroaluminate melt electrolytes, thus opening the research era of AIBs[7]. Since then, positive electrodes and electrolytes have always been the key materials for enhancing the battery performance[8–10].

However, the realization of both high energy density and long cycling life in AIBs is still the bottleneck[9,11]. For one thing, the traditional graphite positive electrode based on chloroaluminate-anions intercalation/deintercalation mechanism are facing a low theoretical

specific capacity due to the limited interlayer space and large size of chloroaluminate anions[12,13]. For another, the conversion-type positive materials, such as S[14,15], Se[16], Te[17] and their intermetallic compounds[18–20], show ultrahigh initial specific capacity and require less aluminum chloride (AlCl$_3$) in terms of storing the given charge (compared with graphite positive electrode) due to their multi-electron transfer conversion mechanism, which provides an opportunity to meet the requirement of high energy density[21]. Unfortunately, most of them are still suffering from poor cycling performance even if the materials are optimized by various methods, including control morphology, composite with conductive materials, etc[22–24]. And it could be attributed to the strong chemical/electrochemical dissolution of conversion-type materials and their intermediate products in the chloroaluminate ionic liquid (IL) electrolytes with strong Lewis acid[25] (Fig. 1a). Subsequently, driven by the electric field and concentration field, the dissolved materials would migrate irreversibly from the positive region towards the negative region (shuttle effect), which not only leads to the loss of active materials, but also blocks the

[1]State Key Laboratory of Advanced Metallurgy, University of Science and Technology Beijing, Beijing 100083, China. [2]School of Metallurgical and Ecological Engineering, University of Science and Technology Beijing, Beijing 100083, China. [3]Institute of Advanced Structural Technology, Beijing Institute of Technology, Beijing 100081, China. ✉e-mail: wwang@ustb.edu.cn; zhewang@ustb.edu.cn; sjiao@ustb.edu.cn

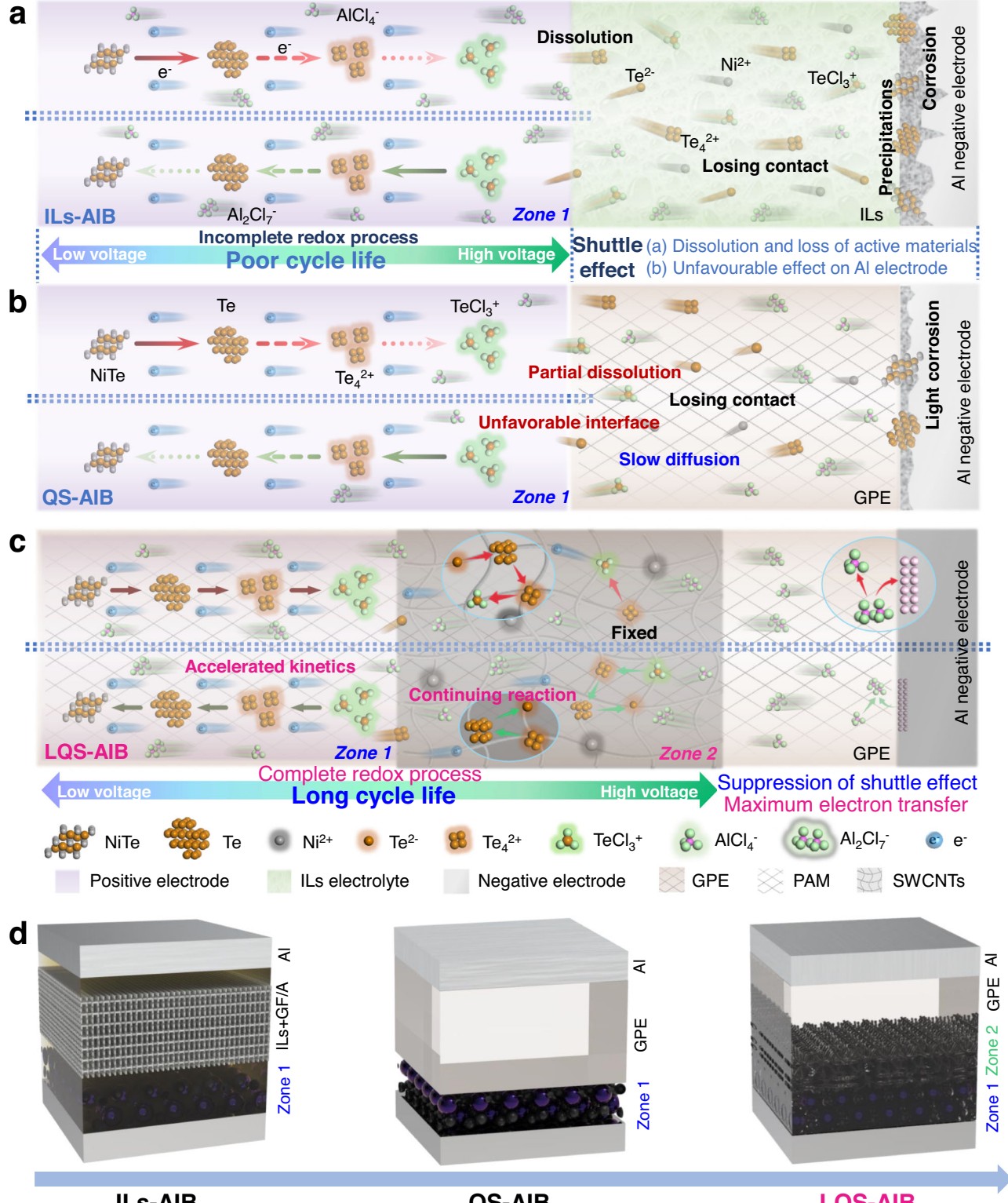

**Fig. 1 | Scheme illustration of mechanism and structure of ILs-, QS- and LQS-AIBs. a, b** Capacity decay mechanism for conversion-type positive materials in ILs-, and QS-AIBs. **c** Working mechanism of LQS-AIB with DRZs for improving electrochemical performance. **d** Structure comparison.

further redox process of Al negative electrode[26]. To make matters worse, the active materials remaining on the positive electrode are also difficult to achieve complete oxidation and reduction due to the polarization caused by the shuttle effect.

Meanwhile, the highly active IL electrolytes could also cause other critical problems, including unexpected gas production and

electrolyte leakage, thus leading to unsatisfied safety and stability of the cell system[27,28]. Therefore, a promising approach is to encapsulate the chloroaluminate ILs within a quasi-solid system to reduce the dissolution and diffusion rate of active materials. Nevertheless, it is still hard to employ the quasi-solid AIB (QS-AIB) systems to effectively enhance the cycling performance of conversion-type positive

electrode materials due to the unfavorable electrode/electrolyte interface and the existing partial dissolution of active materials (Fig. 1b). Therefore, no obvious battery performance improvement is observed for the low utilization of active materials in QS-AIB.

Herein, we have constructed a layered quasi-solid AIB (LQS-AIB) with double micro quasi-solid reaction zones (DRZs) on the positive electrode side via a simple one-step in-situ polymerization method (Fig. 1c, d). The gel precursor solution penetrates into the positive electrode and polymerizes to form the first micro quasi-solid reaction zone (Zone 1), effectively reinforcing the adhesion and wetting ability of the quasi-solid electrolyte to the active materials, thus greatly accelerating the redox reaction kinetics. In the second micro quasi-solid reaction zone (Zone 2), the single-walled carbon nanotubes (SWCNTs) with oxygen-containing functional groups are introduced into the polyacrylamide (PAM)-based gel (C-gel). Noticeably, the SWCNTs serve as both the adsorption additive and electronic conductor to establish a robust three-dimensional (3D) conductive framework with the PAM-based gel in Zone 2, for fixing the active materials dissolved from Zone 1 and simultaneously providing sufficient ion and electron transport channels for further reaction. This particular structure with double micro reaction zones can not only suppress the shuttle effect of the dissolved active materials, but also realize the maximum electron transfer of the positive electrode materials by accelerating reaction kinetics of the active materials in Zone 1 and continuing electrochemical reaction with the active materials dissolved to Zone 2. To prove the practicability of the DRZs structure in LQS-AIB, we take NiTe nanorods positive electrode as an example to assemble the pouch cells. As a result, a high capacity (400 mAh g$^{-1}$) of NiTe positive electrode is maintained after a long-term cycling (4000 cycles). This work opens up a possibility for surpressing the shuttle effect of conversion-type positive materials during cycling and realizing high-capacity and long-life AIBs.

## Results

### Construction of the LQS-AIB with DRZs

The preparation process of DRZs is illustrated (Supplementary Fig. 1) and the PAM-based gel containing active ions ($AlCl_4^-$ and $Al_2Cl_7^-$) is selected for ion transport in the whole quasi-solid AIBs, which is used as both quasi-solid electrolyte and ion conductive agent. Then, a conductive slurry consisting of the gel precursor and SWCNTs was cast onto the positive electrode. As part of the liquid gel precursor penetrated into the positive electrode (Zone 1), an integrated quasi-solid layered structure was constructed after in-situ polymerization with the Zone 2 formed by SWCNTs and remaining gel.

To prove whether the DRZs could suppress the shuttle effect of conversion-type materials to improve its cycling performance, we take NiTe nanorods positive electrode as an example to assemble the pouch cells. The X-ray diffraction (XRD) pattern of the positive electrode material could be assigned to the hexagonal NiTe phase (JCPDS No. 65-9466, Supplementary Fig. 2a) and the field emission scanning electron microscopy (FE-SEM) images suggest its uniform rod-like morphology (Supplementary Fig. 2b, c). Additionally, transmission electron microscopy (TEM) images demonstrate that the nanorods are composed of many black nanoclusters (Supplementary Fig. 2d–f).

It has been confirmed that the NiTe positive electrode would be oxidized to Te, $Te_n(AlCl_4)_2$, and finally $TeCl_3AlCl_4$ during the charging, as well as a reversible reduction process during discharging[20]. However, the capacity decayed rapidly due to dissolution of the active materials (including the original NiTe and intermediate products of Te, $Te_n(AlCl_4)_2$, $TeCl_3AlCl_4$). Here, density functional theory (DFT) calculations were employed to demonstrate the selective adsorption of carbon nanotubes. The SWCNTs with different oxygen-containing functional group (SWCNT, SWCNT-OH, SWCNT-COOH) were taken into account in evaluating adsorption ability for the soluble ions ($Ni^{2+}$, $Te^{2-}$, $Te_4^{2+}$ as an example of $Te_n^{2+}$, $TeCl_3^+$) in the positive electrode and

the major anions ($AlCl_4^-$, $Al_2Cl_7^-$) in the electrolyte. The relaxed structures as well as the calculated total energies of the individual and corresponding adsorption models are collected (Supplementary Figs. 3–5 and Supplementary Table 1). With the introduction of binding energy ($E_b$, see definition in Methods), the adsorption behaviors of SWCNT-OH and SWCNT-COOH are confirmed more favorable for the soluble ions in comparison with SWCNT without oxygen-containing functional groups (Supplementary Fig. 6 and Supplementary Table 2). It is worth noting that oxygen-containing functional groups have little effect on the adsorption of chloroaluminate anions, and the adsorption is relatively weak. Thus, SWCNTs were chemically modified by acid treatment[29] to attach the oxygen-containing functional groups. From the SEM images (Supplementary Fig. 7), the acidified SWCNTs with the rough outer wall and a length of tens of micrometers have better dispersion. And the metal impurity elements in the original SWCNTs are removed, and the oxygen content increases obviously (Supplementary Fig. 7c, d). In the X-ray photoelectron spectroscopy (XPS) survey scan of the SWCNTs samples (Supplementary Fig. 8a), pronounced peaks corresponding to C 1$s$ and O 1$s$ transition are observed for the acidified SWCNTs. The results of C 1$s$ XPS spectra (Supplementary Fig. 8b, c) reveal the presence of peaks for SWCNTs corresponding to sp$^2$-hybridized carbon (284.8 eV), sp$^3$-hybridized carbon (285.1 eV), hydroxyl carbon (286.6 eV), carboxyl carbon (288.9 eV) and $\pi$-$\pi$* transitions at higher binding energy range (290.9 eV) in different relative percentages of each contribution before and after acidification[30,31] (Supplementary Table 3). And the carboxyl and hydroxyl functional groups on acidified SWCNTs are also confirmed through the deconvolution of XPS O 1$s$ peak (Supplementary Fig. 8d). The enhanced $I_D/I_G$ ratio in Raman spectra (Supplementary Fig. 9a) also indicates the successful functionalization of the acidified SWCNTs[32]. Furthermore, the acidified SWCNTs reveal a negative zeta potential of −15 mV (Supplementary Fig. 9b), which is highly beneficial to the adsorption of cations (such as $Ni^{2+}$, $Te_n^{2+}$, and $TeCl_3^+$) dissolved in electrolyte.

Meanwhile, the robust adsorption behaviors of SWCNT-OH and -COOH on soluble active ions are more intuitively demonstrated by the results of charge density difference contours (Fig. 2a). It can be found that there are a large amount of charge accumulation (denoted by a green cloud) and depletion (denoted by a yellow cloud) residing at the contact edge between SWCNT-OH (or -COOH) and the soluble active ions ($Ni^{2+}$, $Te^{2-}$, $Te_4^{2+}$, $TeCl_3^+$). The total density of states (TDOS) of SWCNT-OH and -COOH matrices and the partial density of states (PDOS) of the adsorption models are calculated (Fig. 2b, c), and strong electron contributions from adsorbed ions are detected near the Fermi level (denoted by the vertical black dashed line). It also proves that the strong absorption behaviors between SWCNTs (-OH, -COOH) and the soluble active ions are achieved via electron interactions. The optimum content of acidified SWCNTs dispersed in the conductive slurry was investigated by galvanostatic charge and discharge tests. The LQS-AIB exhibits better cycling performance when the content of SWCNTs increases to 1 wt% (mass percentage to gel precursor, Supplementary Fig. 10). Although more SWCNTs are beneficial for capturing the dissolved active ions and enhancing the electronic conductivity of Zone 2, the increased viscosity also hinders the ions transport. Therefore, the SWCNTs content of 1 wt% is applied in the following tests.

The morphologies and structure of the LQS-AIB were characterized by FE-SEM techniques. In order to gain clear image information, the cell cores were immersed in dichloromethane (DCM) solvent for 6 h to remove part of the PAM-based gel. The SEM image for the cross-section view of LQS-AIB is shown in Fig. 2d and each part of the quasi-solid battery is closely connected. It could also be observed from the cracks generated after immersion that the gel is well infiltrated and interwoven in the positive electrode materials in Zone 1 (Fig. 2e and Supplementary

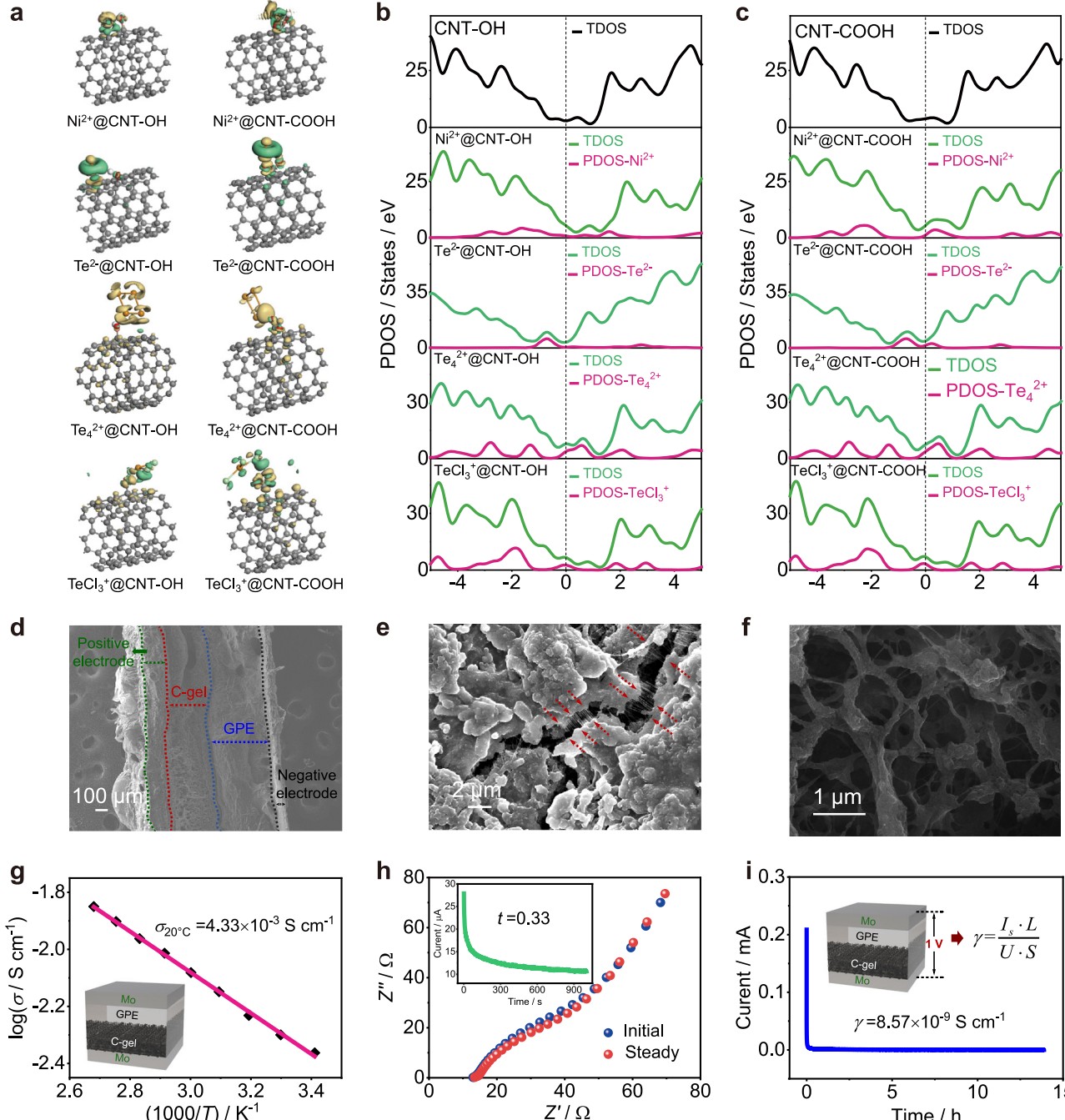

**Fig. 2 | Construction of the LQS-AIB with DRZs. a** Charge density difference of SWCNT-OH (or -COOH) and soluble active ions. Green clouds indicate the charge accumulation, and yellow clouds the charge depletion. Carbon, hydrogen, oxygen, nickel, tellurium, chlorine atoms are denoted using gray, white, red, blue, brown, and light green spheres, respectively. **b**, **c** TDOS and PDOS of SWCNT-OH and -COOH matrices with the adsorption models. **d** SEM image for the cross-section view of LQS-AIB. **e** SEM image of Zone 1. **f** SEM image of Zone 2 (C-gel) after soaking treatment. **g** Arrhenius conductivity plot determined by AC impedance spectro-scopy experiments. **h** Transference number ($t$.) measured in the symmetric cell. **i** Electronic conductivity ($\gamma$) of the C-gel/GPE.

Fig. 11). The SEM images of the C-gel before and after soaking treatment are compared (Fig. 2f and Supplementary Fig. 12) and it shows that the surface of the polymerized C-gel is smooth and uniform, while the 3D network structure is exposed after removing part of gel in Zone 2. The ion transport capacity of the C-gel/GPE was measured and analyzed by electrochemical impedance spectroscopy (EIS) method. The ionic conductivities are determined to be $4.33 \times 10^{-3}$ and $1.41 \times 10^{-2}$ S cm$^{-1}$ at 20 and 100 °C (Supplementary Fig. 13), respectively, the values of which are approximate to the reported pure gel[33]. The plot of log conductivity versus 1000/$T$ exhibits a linear relationship, suggesting

they follow the Arrhenius behavior (Fig. 2g). By calculating the slope, the activation energy ($E_a$) here is evaluated to be 16.33 eV, indicating a fast ion transport[34]. In addition, the anion transfer-ence number ($t$.) of the C-gel/GPE is 0.33 (Fig. 2h, Supplementary Fig. 14), higher than that of ILs[28], which is beneficial to improve electrochemical kinetics and reduce concentration polarization. These indicate that the adsorption effect of 1 wt% SWCNTs on anion conduction could be almost ignored. The low electronic conductivity ($8.57 \times 10^{-9}$ S cm$^{-1}$) of the C-gel/GPE was obtained from the polarization current-time plot (Fig. 2i), which meets the basic requirements of electrochemical devices[35].

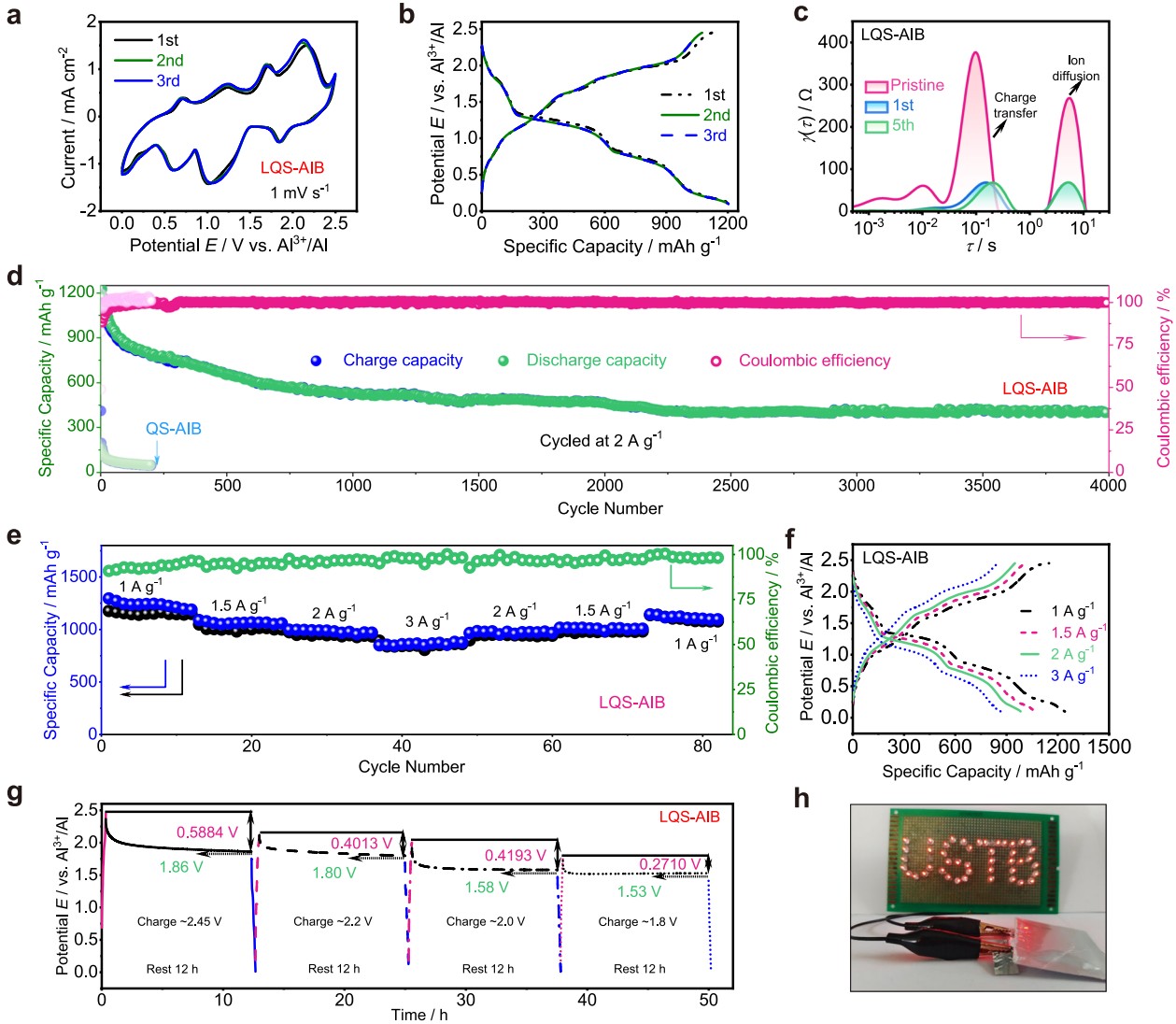

**Fig. 3 | Electrochemical performance. a** CV curves of LQS-AIB at a scan rate of 1 mV s⁻¹. **b** Initial charge and discharge behaviors of LQS-AIB at 2 A g⁻¹. **c** DRT curves of the EIS for different cycles of LQS-AIB. **d** Comparison of long-term stability tests of LQS- and QS-AIBs at 2 A g⁻¹. **e** Rate capability of LQS-AIB at various current densities from 1 to 3 A g⁻¹. **f** Corresponding galvanostatic charge and discharge profile at different current densities. **g** Self-discharge behavior for different charge states of LQS-AIB. **h** One fully charged LQS-AIB powering up a lamp panel (working voltage: 1.8 V).

## Electrochemical performance of the LQS-AIBs

The initial open-circuit potential of an assembled LQS-AIB was measured as approximately 1.42 V (Supplementary Fig. 15). Then the typical cyclic voltammetry (CV) measurements were performed within the voltage range of 0-2.5 V at a scan rate of 1 mV s⁻¹ (Fig. 3a). And the CV curves are well overlapped, representing a superior reversibility. Four redox pairs are located at 2.13/1.84 V, 1.69/1.05 V, 1.23/0.62 V and 0.69/0.28 V for LQS-AIB, respectively, corresponding to the complex conversion processes of $Te^{2-} \leftrightarrow Te \leftrightarrow Te_n^{2+}$ ($1 \le n \le 8$) $\leftrightarrow TeCl_3^+$. Herein, $Te_n^{2+}$ exists in the form of polycationic clusters, such as $Te_4^{2+}$、$Te_6^{2+}$, and $Te_8^{2+}$, which are stabilized by the large anions in Lewis acidic melts[36–38]. When the battery was charged to 1.5 V, the oxidation product was confirmed to be $Te_4(AlCl_4)_2$ in our previous work[20]. Therefore, it can be inferred that during the charging process, $Te^{2-}$ is first converted to Te, and further oxidized to form long-chain $Te_n^{2+}$ ($4 < n \le 8$), which is then converted to short-chain $Te_n^{2+}$ ($1 < n \le 4$) to generate $Te^{2+}$, finally being oxidized to $TeCl_3^+$ under high potential. In the subsequent discharge process, the corresponding reversible reduction reactions occur. In the case of QS-AIB, the CV curves reveal the absence of reduction peak A' and

oxidation peak D (Supplementary Fig. 16). It could be concluded that the reaction products formed at high or low potentials are more prone to be dissolved in the acidic electrolyte and get permanently isolated from the current collector. As a result, these dissolved products are unable to continue participating in the electrochemical reaction. Additionally, all peak currents gradually decrease with cycling, further confirming the dissolution of the active material in QS-AIB.

The galvanostatic charge and discharge tests were conducted subsequently. Fig. 3b demonstrates the initial charge and discharge behaviors of the LQS-AIB at a current density of 2 A g⁻¹. It could be observed that four pairs of charge and discharge voltage plateaus appear at 2.13/1.84 V, 1.69/1.05 V, 1.23/0.62 V, and 0.69/0.28 V, respectively, matching well with the CV curves. Particularly, the almost overlapped charge and discharge curves also reveal good stability of the LQS-AIB. To evaluate the capacity contribution of the SWCNTs in Zone 2, the batteries without NiTe material were fabricated in the same way as LQS-AIB and no redox peak is observed in the CV curves (Supplementary Fig. 17a). Meanwhile, low capacities of less than 20 mAh g⁻¹ at a current density of 2 A g⁻¹ are obtained (Supplementary

Fig. 17b, c), demonstrating the negligible capacity contribution to the total capacity of LQS-AIB.

EIS technique was used to investigate the kinetic behavior in the charge and discharge processes (Supplementary Fig. 18). Each Nyquist plot is composed of two depressed semicircles at high frequencies and a straight line at low frequencies, corresponding to the interfacial charge transfer resistance ($R_{ct}$) and Warburg resistance ($Z_w$), respectively. By contrast, the total resistance of LQS-AIB is sharply reduced to 121 Ω after the 1st cycle and then remained at 105 Ω after 5 cycles, indicating a gradual process for battery activation. The corresponding distribution of relaxation times (DRT) analysis[39] also confirms that the impedance of electrochemical reaction and ion diffusion both show rapid attenuation after cycling (Fig. 3c), resulting from the formation of the more stable C-gel/GPE interface. The evolution process of gel interface could be associated with the chemical reaction and decomposition of the passivation layer formed by incompletely polymerized monomer. Thus, the kinetics of electrochemical reaction would be improved during the first few cycles.

As shown in Fig. 3d, the cycling performances of LQS- and QS-AIBs are evaluated and compared at the current density of 2 A g$^{-1}$. Apparently, the LQS-AIB exhibits more excellent reversibility and better cycling stability up to 4000 cycles with a high remaining capacity of ~400 mAh g$^{-1}$, while there is rapid capacity decay in the QS-AIB. Additionally, the initial capacity of QS-AIB is also much lower than that of LQS-AIB. This is primarily attributed to the inadequate wetting ability of quasi-solid electrolyte to NiTe positive electrode and partial dissolution of the original active materials. These factors hinder the complete involvement of the active materials in the electron transfer process during both charge and discharge stages, ultimately leading to a low utilization rate of active materials. In contrast, these problems are addressed in LQS-AIB by constructing the DRZs, improving both the capacity and stability of the battery. Supplementary Fig. 19 shows the corresponding charge and discharge curves at different cycles (20th-1000th) of the LQS-AIB. The obvious charge and discharge voltage plateaus are maintained throughout the cycles with Coulombic efficiencies of nearly 100%. In order to further compare the enhancing effect of the DRZs on battery performance, the cycling stability was also tested for AIB with modified separator (MS-AIB) based on the same SWCNTs (Supplementary Fig. 20). It is obvious that the electrochemical performance of MS-AIB is much inferior to that based on DRZs (Supplementary Fig. 21). This further highlights the effectiveness of constructing the DRZs to form the LQS-AIB as an optimization strategy.

Furthermore, the rate capability of the LQS-AIB was tested at different current densities from 1 to 3 A g$^{-1}$. The average discharge specific capacities remain very steady of 1241, 1063, 982, and 865 mAh g$^{-1}$ at 1, 1.5, 2, and 3 A g$^{-1}$, respectively, demonstrating a favorable rate capability in the LQS-AIB (Fig. 3e). Meanwhile, the rate capability curves approach the initial ones when cycled at the same rate, indicating high reversibility and cycling stability of the LQS-AIB based on the DRZs. And the corresponding charge and discharge curves are shown in Fig. 3f at the current densities from 1 to 3 A g$^{-1}$. Additionally, the self-discharge behavior, one of the parameters to judge battery performance, was further studied for different charge states of the LQS- and QS-AIB (Fig. 3g and Supplementary Fig. 22). The batteries were separately charged to 2.45, 2.2, 2.0, and 1.8 V, followed by settling for 12 h, and then the voltage drop values were recorded. The self-discharge ratios of the LQS-AIB are much smaller than that of the QS-AIB at any charging state and the stable open-circuit potential values of >1.5 V are always maintained, indicating that the constructed quasi-solid DRZs can fix the dissolved active materials to suppress their shuttle effect in LQS-AIB to maintain the working stability of battery devices. In addition, it is observed in Fig. 3h that one fully charged LQS-AIB could power up a lamp panel consisting of tens of light-emitting-diodes (LEDs, working voltage: 1.8 V) in parallel.

## Stable interface in the layered structure of LQS-AIB

To investigate the internal structure of the reaction zones in the positive electrode, ex-situ X-ray nano-computed tomography (CT) technique was employed for the QS- and LQS-AIB systems (Fig. 4a–h). As shown in Fig. 4a and e, the real internal structure was reconstructed by collecting CT image information between positive electrode and the quasi-solid gel. And the corresponding profiles of the two battery systems from front, left and top view are displayed in Fig. 4b–d and f–h, respectively. Obviously, it can be observed that there are many narrow gaps and holes at the interface between the positive electrode and GPE of the QS-AIB, which is unfavorable for the transport of active anions to positive electrode materials. In contrast, the positive electrode penetrated by gel and C-gel on its surface form an integral quasi-solid structure with two layers in LQS-AIB through in-situ polymerization, showing a tight and robust positive electrode/C-gel interface and enhanced adhesive strength between the current collector and the positive electrode materials. The C-gel/GPE interface structure was also characterized and shows the same compact structure, indicating that the two gels are well combined owing to the similar composition (Supplementary Fig. 23).

Furthermore, the EIS and the corresponding DRT analysis were applied to study the reaction dynamics in the QS- and LQS-AIBs. According to the Nyquist plots and equivalent circuit model (Fig. 4i, Supplementary Fig. 24), the value of $R_{ct}$ is decreased with the reduction of the second semicircle when the DRZs is employed in the LQS-AIB. The applicability of the equivalent circuit model has been verified by Chi-square values, which are lower than 10$^{-3}$ (Supplementary Table 4), representing good fitting quality. The DRT results from the EIS data are shown in Fig. 4j. The P1, P2, and P3 time regions represent the contact resistance, electrochemical reaction resistance, and ion diffusion resistance, respectively[40]. Apparently, the relaxation time of ion diffusion has decreased in LQS-AIB, which could be ascribed to the enhanced interfacial contact in the internal battery system. Therefore, the robust and stable interface in the layered structure would accelerate the anion transport kinetics between positive electrode materials and gel electrolyte, resulting in the reduced electrochemical reaction impedance.

Significantly, the cycling performance of LQS-AIB is competitive against the record-setting values of many other battery systems based on conversion-type positive electrode materials[20,23,24,26,41–43] (Fig. 4k and Supplementary Table 5). And combined with the relation of specific capacity versus discharge voltage, it could be demonstrated that the construction of DRZs is an effective strategy to develop stable high-energy-density AIBs with conversion-type positive electrodes. In addition, the positive electrode materials based on other different working mechanism[44–47], including intercalation/deintercalation of chloroaluminate ions, alternate storage of opposite charges in multisites, etc., were also used for comparison (Supplementary Table 5). Although these positive electrode materials can achieve relatively long cycling life, the lower energy density they maintain is a limiting factor for practical applications. Apparently, LQS-AIB not only presented an improved long-term cycling performance, but also delivered high energy density. Therefore, it is very significant to develop LQS-AIB with DRZs for conversion-type positive electrodes to realize high capacity and long-term stability. Meanwhile, the ultrahigh safety of quasi-solid battery in LQS-AIB is also favorable in practical application.

## Suppression of shuttle effect

The CV curve of LQS-AIB after 200 cycles exhibit a similar shape to the initial one in Fig. 3a, while that of QS-AIB almost has no obvious redox peak (Fig. 5a). The charge and discharge curves are shown in Fig. 5b for the two battery systems at the 20th and 200th cycles. The four pairs of charge and discharge voltage plateaus and the smaller electrochemical polarization (0.68 vs. 1.18 V and 0.7 vs. 1.06 mV at 20th

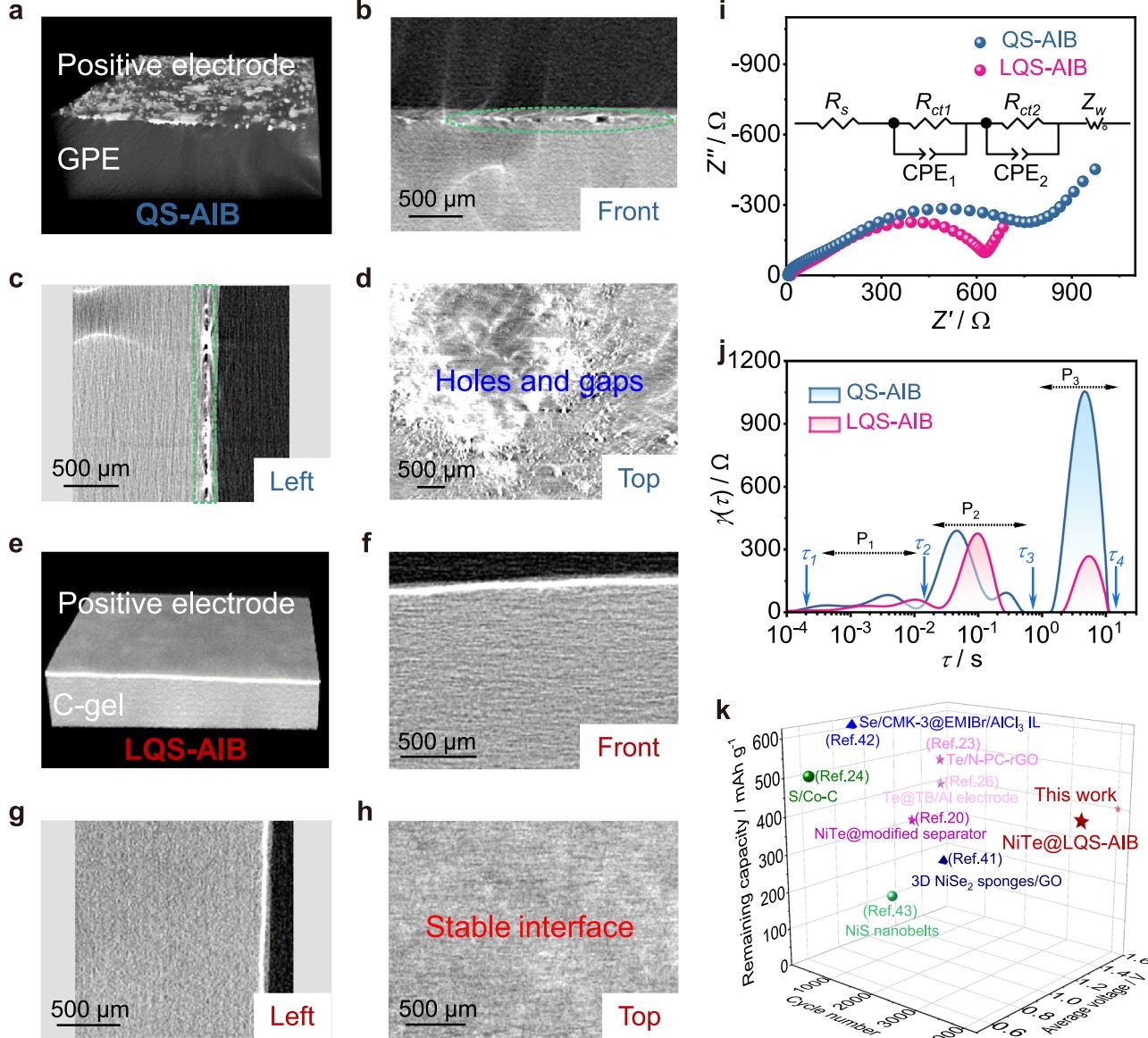

**Fig. 4 | Stable interface in the layered structure of LQS-AIB.** Ex-situ X-ray nano-computed tomography (CT) images of (**a**) reconstructed architecture of Zone 1 in QS-AIB and (**b**) front, (**c**) left and (**d**) top view of the corresponding profiles. **e** Reconstructed architecture of the DRZs in LQS-AIB and (**f**) front, (**g**) left and (**h**) top view of the corresponding profiles of DRZs. **i** Comparison of Nyquist plots for the LQS- and QS-AIBs. Inset shows the equivalent circuit model for the battery systems. **j** DRT results of LQS- and QS-AIBs. **k** Comparison of the performance of our LQS-AIB with the other Al batteries based on conversion-type positive electrode materials.

and 200th cycle for LQS- vs. QS-AIB, respectively) could always be observed in the LQS-AIB due to the favorable redox kinetics. However, unobvious voltage plateaus and severe polarization appear in QS-AIB, caused by the dissolution of the active materials and increased internal resistance. In addition, the electrochemical performance of the ILs-AIB after cycling, similar to that of QS-AIB, was also recorded (Supplementary Fig. 25). Meanwhile, it could be observed from Supplementary Fig. 26 that black precipitates accumulate on the Al negative electrode side with cycling, accompanied by serious electrode corrosion. To understand the chemical components of the black precipitates, energy dispersive X-ray spectroscopy (EDS) and XPS characterizations were conducted. In Supplementary Fig. 27a and b, the Te and Ni signals are obviously detected between GF/A separator and Al foil. Combined with XPS results (Supplementary Fig. 27c, point 2), the black precipitates are proved to be a mixture of NiTe, Te, and tetravalent Te compounds, indicating that the active materials of positive electrode would shuttle through the GF/A separator and deposit on Al negative electrode. Due to the

polarization caused by shuttle effect of dissolved active materials, the ILs-AIBs reach charge and discharge cut-off potential rapidly with poor cycling stability. Therefore, the active materials remaining on positive electrode undergoes an incomplete electrochemical redox process (Supplementary Fig. 27c, point 1, charged to 2.45 V), which dramatically reduces the charge capacity.

To further demonstrate how the two quasi-solid-state AIB systems suppress the shuttle effect, element information is detected and compared at various positions inside cells (Fig. 5c, f) after cycles under charge and discharge states. As can be seen from the Te $3d$ XPS patterns of QS-AIB charged to 2.45 V (Fig. 5d), two obvious Te$^0$ peaks at 573.1 and 583.6 cm$^{-1}$ are ascribed to the incomplete transformation of tellurium intermediate product at point 1 (positive electrode) and the peaks of Te$^{2-}$ and Te$^{4+}$ are also shown at point 2 (GPE, near the Al negative electrode), indicating serious shuttle effect in the QS-AIB. The EDS mapping also reveals obvious Te and Ni signals at point 2 (Fig. 5e), agreeing well with the XPS results. For the fully charged LQS-AIB, the XPS spectra at both point 1 (in Zone 1) and point 1′ (in Zone 2) show two

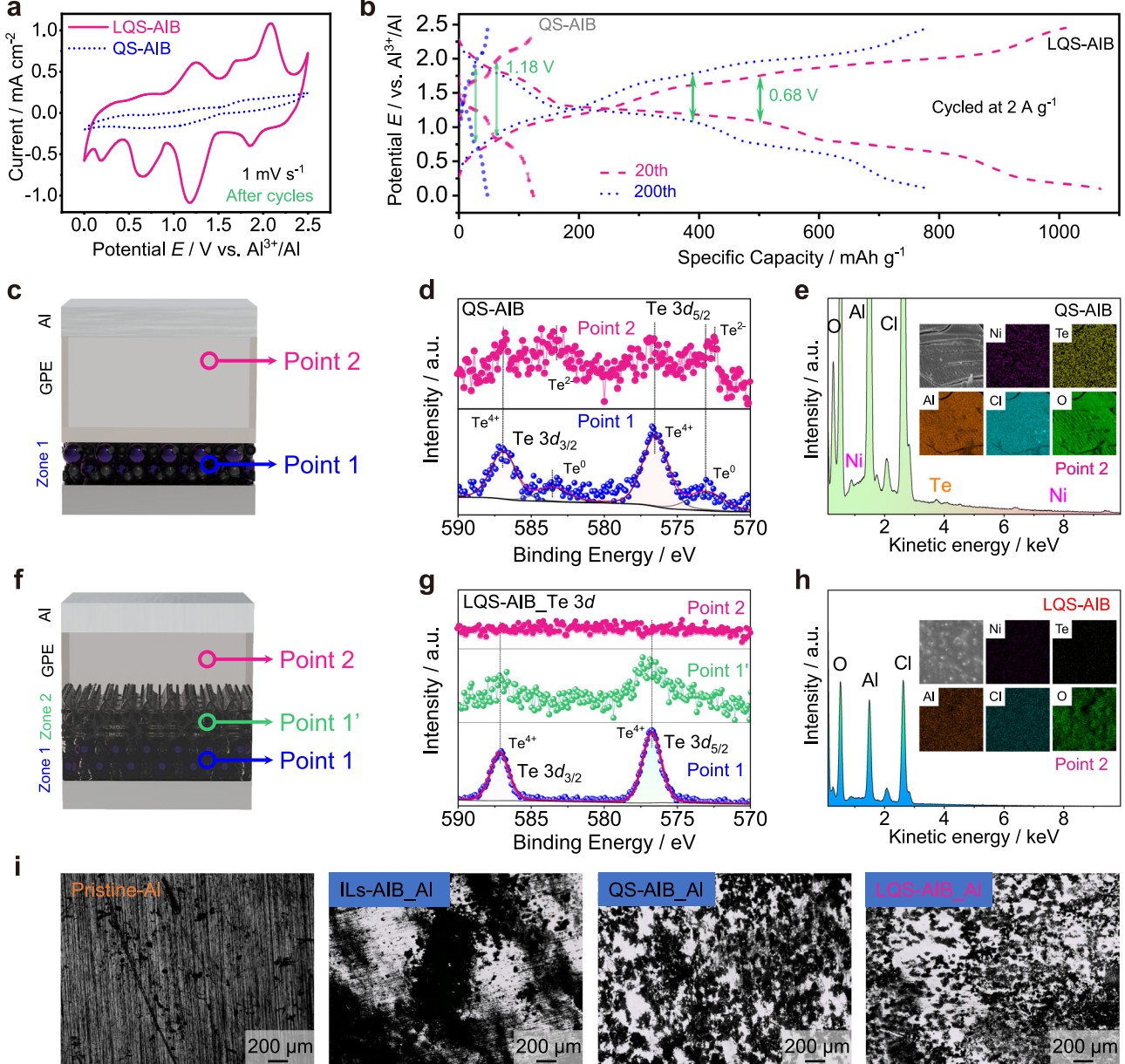

**Fig. 5 | Suppression of shuttle effect. a** CV curves of the LQS- and QS-AIB after 200 cycles. **b** Charge and discharge curves for the two battery systems at the 20th and 200th cycles. **c** Schematic diagram of the different points in the cycled QS-AIB and **d** corresponding Te $3d$ XPS spectra and **e** EDS spectra at point 2. **f** Schematic diagram of the different points in the cycled LQS-AIB and **g** corresponding Te $3d$ XPS spectra and **h** EDS spectra at point 2. **i** Optical images of Al negative electrodes before and after cycles in the ILs-, QS- and LQS-AIBs.

peaks at 576.6 and 586.9 eV referring to $Te^{4+}$ $3d_{5/2}$ and $Te^{4+}$ $3d_{3/2}$, respectively, achieving a complete oxidation of active materials (Fig. 5g). Moreover, no Te and Ni signals are observed at point 2 (GPE, near the Al negative electrode) in the LQS-AIB (Fig. 5h), further confirming that the as-constructed layered DRZs of can effectively suppress the shuttling of dissolved active materials. The comparative results of Te $3d$ patterns at fully discharged state for the positive electrodes of three types of AIBs are also demonstrated in Supplementary Fig. 28. Apparently, the active materials in both IL- and QS-AIBs are still in the states of oxidation or intermediate, while that in LQS-AIB is reduced to the lowest valence ($Te^{2-}$). Therefore, it can be concluded that the DRZs in LQS-AIB can realize the maximum electron transfer of active materials and efficiently suppress the shuttle effect during long-term cycling. To further verify the role of DRZs on suppressing shuttle effect, permeation experiments were conducted using the H-type cells for visualization (Supplementary Fig. 29), in which the

IL electrolyte with dissolved NiTe active materials (left chamber) and the blank IL electrolyte (right chamber) were separated by GF/A separator, GPE and C-gel/GPE, respectively, to simulate the diffusion process within the ILs-AIB, QS-AIB and LQS-AIB. The results after settling for a period of time indicate that the purple dissolved materials can easily pass through the GF/A separator within 1 minute. Consequently, the blank electrolyte in the right chamber darkens in color after only 12 h. And the diffusion rate of purple dissolved materials is slowed down when employing GPE. For the case of C-gel/GPE, almost no visible change in color of blank IL electrolyte is observed even after 36 h, indicating excellent performance of LQS-AIB in suppressing shuttle effects.

Furthermore, Fig. 5i shows the optical images of Al negative electrodes before and after cycling in the ILs-, QS-, and LQS-AIBs. The observation results in different AIB systems suggest the unevenly distributed deposition on the surface of Al electrode, indicating that

the Al dendrites preferentially nucleate and grow at certain regions. And large corrosion pits are shown on the Al foil of ILs-AIB, resulting in the fragmentation of the negative electrode (Supplementary Fig. 26b), which may be caused by conversion reaction between Al and the diffused tellurium species from positive electrode. To verify this, the Al foil was put into the ionic liquid containing dissolved active materials. After 12 h, it can be seen that the purple solution becomes clear and there is black precipitation on the surface of Al foil (Supplementary Fig. 30a). The EDS and XRD results show that the precipitation is a mixture of NiTe and Te (Supplementary Fig. 30b–d), which is consistent with the composition of black powder accumulated between GF/A separator and Al negative electrode (Supplementary Fig. 26b) during cycling, suggesting that the chemical reaction can be inferred as:

$$3TeCl_4 + 4Al \rightarrow 3Te + 4AlCl_3 \qquad (1)$$

$$TeCl_4 + 2Al + NiCl_2 \rightarrow NiTe + 2AlCl_3 \qquad (2)$$

In contrast, the corrosion of Al negative electrode in QS-AIB is relatively slight due to the slow diffusion of dissolved materials in quasi-solid-state system. Notably, the surface of Al negative electrode in LQS-AIB after cycling is relatively uniform, and no Ni and Te signals are detected (Supplementary Fig. 31). It is again proved that the DRZs in LQS-AIB can significantly suppress the shuttle effect and further inhibit the possible corrosion and side reactions on Al negative electrode. Therefore, the reversible redox reaction based on the maximum electron transfer (six-electron transfer of tellurium species) is achieved by constructing the DRZs for long-life aluminum-ion batteries.

## Discussion

In summary, an LQS-AIB has been established with double micro quasi-solid reaction zones to improve electrochemical performance of the conversion-type positive electrode. The redox reaction kinetics are accelerated in Zone 1 with enhanced wetting ability and interfacial adhesion. Zone 2 provides physical space to confine the active materials dissolved from Zone 1 and to continue the uncompleted redox reaction of these active materials, thus suppressing the shuttle effect. It also bridges the electronic and ionic channels between Zone 1 and GPE for fast electrochemical reaction. Consequently, the LQS-AIB exhibits a high specific capacity of 400 mAh g$^{-1}$ after a long-term cycling (4000 cycles), superior to the recently reported AIBs. This attempt demonstrates that a stable and reversible electron transfer process can be achieved during long-term cycling for conversion-type positive electrode of AIBs. Most notably, it can be concluded that, even when considering the mass of the electrolyte, the conversion reactions with multi-electron transfer of conversion-type positive electrode such as NiTe are more competitive in energy density due to the less AlCl$_3$ required for the given charge storage (See details in Supplementary Note), compared with typical graphite or organic positive electrodes[47,48]. Accordingly, the design of DRZs to construct the LQS-AIBs could be considered as an effective way to address the existing challenge of balancing high energy density and long cycle life in aluminum-ion batteries.

## Methods

### Materials

Na$_2$TeO$_3$ (0.005 mol, purity 99.99%, J&K), NiCl$_2$·6H$_2$O (0.005 mol, purity 98%, Aladdin), N$_2$H$_4$·H$_2$O (80%, Energy Chemical), Acrylamide (98.5%, Alfa Aesar), dichloromethane (DCM, 99 + %, Alfa Aesar), 1-ethyl-3-methylimidazolium chloride ([EMIm]Cl, 97%, Acros Chemicals), aluminum chloride (AlCl$_3$, anhydrous, 99.999%, Sigma Aldrich), single-walled carbon nanotubes (SWCNTs, XFNANO), 2, 2'-azodiisobutyronitrile (initiator AIBN, >98%, TCI), were obtained from commercial sources. Aluminum foil (50 μm thickness, 99.99%), molybdenum foil (20 μm thickness, 99.0%) were received from General Research Institute for Nonferrous Metals (Beijing, P.R. China). Tantalum foil (10 μm thickness, 99.9%) was purchased from Sheng Yuan metal co. LTD.

### Preparation of the NiTe nanorod-positive electrodes

In a typical synthesis, 1.2 g of NiCl$_2$·6H$_2$O and 1.1 g Na$_2$TeO$_3$ were dissolved in 50 mL deionized water under sonication. With continuously stirring, 20 mL N$_2$H$_4$·H$_2$O was then slowly added to the above solution. Subsequently, the obtained suspension was transferred into a 100 mL autoclave and treated at 140 °C for 6 h. After cooling to room temperature, the precipitate was washed with distilled water and dried in vacuum to obtain NiTe nanorod powder. Then the NiTe positive electrodes were prepared by coating with a slurry formed by mixing 70 wt% NiTe powder, 20 wt% acetylene black, 10 wt% binder (CMC), and a proportion of solvent (deionized water) onto the Ta foil. And the average loading amount was around 1 mg cm$^{-2}$.

### Preparation of the acidified SWCNTs

The purified SWCNTs were suspended in a 3:1 v/v solution of concentrated sulfuric acid (98%) and concentrated nitric acid (70%), and ultrasonicated for 2 h. Then the product was repeatedly washed with deionized water and dried in vacuum at 60 °C.

### The construction of the DRZs

The anhydrous AlCl$_3$ and [EMIm]Cl were mixed with a molar ratio of 1.3:1 under constantly stirring in an argon-atmosphere glovebox, followed by standing for at least 24 h to form a homogeneous ionic liquid (IL) plasticizer. AlCl$_3$ (0.02 mol) and acrylamide (0.01 mol, monomer of polymer) were slowly dissolved into the as-prepared IL plasticizer solution (8 g) with stirring to obtain a transparent yellow solution. Different amounts (0.5 wt%, 0.8 wt%, and 1 wt%) of the acidified SWCNTs were pre-dispersed in DCM by high-power ultrasonication for 24 h and then mixed with above solution through magnetic stirring. After adding the initiator AIBN (1 wt% of monomer), the slurry composed of gel precursor and SWCNTs was coated onto the positive electrodes. During a settling time for 12 h, part of the liquid gel precursor penetrated into the positive electrode (Zone 1), and then the acrylamide monomers were in-situ polymerized to form an integrated quasi-solid DRZs with the layered structure including Zone 1 and Zone 2. And Zone 2 was composed of SWCNTs and remaining gel to form a C-gel on the suface of Zone 1.

### Assembly of the LQS-AIBs

To obtain the gel polymer electrolyte (GPE) on negative side, the precursor solution was prepared by the same procedure except for the addition of SWCNTs and then cast onto the Al negative electrode. A piece of Mo foil was attached onto the blank side of the Ta foil as the positive electrode tab and the negative electrode tab still used Al foil. With the integrated quasi-solid positive electrode and GPE on Al negative electrodes, the pouch cells were assembled into the outer package made by Al-plastic film and sealed through the overwrapping machine (BLEUETS FR-300B, sealing temperature below 300 °C under atmospheric pressure). In comparison, the blank batteries were fabricated in the same way without NiTe active materials to determine the contribution to the specific capacity of the SWCNTs in gel.

### Electrochemical measurements

The galvanostatic discharge and charge tests were performed by a multichannel battery testing system (Neware BTS-53) at various current densities. The Cyclic voltammetry (CV) measurements were implemented by CHI 1140 C electrochemical workstation (Shanghai, China) at different scan rates. Electrochemical impedance spectroscopy (EIS) was carried out with a CHI 660E electrochemical workstation (Shanghai, China) at the frequency range of 100 kHz–0.01 Hz

with oscillation amplitude of 5 mV under the open circuit potential. And the transition for distribution of relaxation times (DRT) was conducted using RelaxIS impedance spectrum analysis.

The ionic conductivities were analyzed by EIS method with two blocking Mo electrodes under the working temperatures from 20 ~ 100 °C. The transference number ($t$.) was measured by the steady-state technique which involves a combination of AC and DC measurements with two Al foils as nonblocking electrodes. The AC impedance measurement was applied to a frequency range from 100 kHz to 0.01 Hz with oscillation amplitude of 5 mV and then a small voltage pulse ($\Delta V = 10$ mV) was applied for the following DC polarization. The electrical conductivity was obtained from a polarization current-time plot and the C-gel/GPE was sandwiched between two Mo electrodes. Under 1 V polarization voltage, the symmetric Mo electrode would block the $EMI^+$ and $Al_xCl_y^-$ ions; only electronic flow remains constant during the whole sample cell. The electrical conductivity ($\gamma$) could be determined by:

$$\gamma = \frac{I_s \cdot L}{U \cdot S} \quad (3)$$

where $I_s$ is the stead-state current, $U$ the polarization voltage, $L$ and $S$ the thickness and the surface area of the C-gel/GPE, respectively.

## Materials characterizations

The morphology and element distributions of the samples were characterized by FE-SEM (JEOL, JSM6701F) with an energy-dispersive X-ray spectrometer and transmission electron microscope (TEM, JEOL, JSM2100F). The valence states of the element were detected on XPS (Kratos AXIS Ultra DLD) spectra. XRD analysis was carried out by a Rigaku/Smartlab Instrument (Cu Kα, 45 kV, 200 mA) at a scan speed of 10°/min to determine the crystal structures with the $2\theta$ range of 10-90°. The X-ray nano-computed tomography (CT, MicroXCT-400) technique was performed to investigate the internal structure of the LQS-AIB. Raman spectra were collected on a LabRAM HR Evolution spectrometer, with a laser wave wavelength of 532 nm and a power of 5 mW. Zeta potential tester (Malvern Zetasizer Nano ZS ZEN3600) was used to study the charge potential of SWCNTs.

## Theoretical calculation

Systematic density functional theory (DFT) calculations were performed in the generalized gradient approximation (GGA) with Perdew–Burke–Ernzerhof (PBE) functional for exchange and correlation effects of the electrons, as implemented in the DMol3 package[49,50]. The Brillouin-zone integration was sampled by $5 \times 5 \times 5$ special k-points for atomic structure relaxation. The global orbital cut-off energy for each element was set to 4.5 Å. Geometry optimization was carried out with convergence to a self-consistent-field (SCF) tolerance of $1 \times 10^{-5}$ Ha, a maximum force of 0.002 Ha Å$^{-1}$ and a maximum displacement of 0.005 Å. The choice of these computational parameters ensured good convergence in the present studies.

In order to quantitatively investigate the anchoring effect (chemisorption) of SWCNT (-OH, -COOH) to soluble ions, binding energy ($E_b$) was defined as follows:

$$E_b = E_{composite} - E_{SWCNT} - E_{ion} \quad (4)$$

where $E_{composite}$ represents the total energy of SWCNT (-OH, -COOH) combining with soluble ions, $E_{SWCNT}$ the total energy of SWCNT (-OH, -COOH) and $E_{ion}$ the total energy of the soluble ion. Thus, the greater the more negative value of $E_b$, the stronger binding of soluble ions to SWCNT (-OH, -COOH) under this definition.

## Data availability

The data generated in this study are provided in the Source Data file. All additional information is available from the corresponding authors upon request. Source data are provided with this paper.

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

## Acknowledgements

This work was supported by National Natural Science Foundation of China (Grant No. 52104294 to Z. Y., Grant No. 51725401 to S. J.), the China Postdoctoral Science Foundation (Grant No. 2021M700397 to Z. Y.) and the Beijing Nova Program (Grant No. Z211100002121082 to W. W.). The authors also thank Xiaomi Foundation for the financial support of Xiaomi Young Scholar Program.

## Author contributions

S. J. and W. W. conceived the project. S. J. and Z. W. designed the experiments. Z. Y. carried out the experiments. Z. Y. and W. W. co-wrote the manuscript. Y. Z., W. -L. S. and Z. H. participated in characterizations and result analysis. All authors discussed the results.

## Competing interests

The authors declare no competing interests.
