## [Peer Review File · Nature Communications]

REVIEWER COMMENTS

Reviewer #1 (Remarks to the Author):

This manuscript describes a layered quasi-solid structure for a high-capacity conversion-type cathode. The layered structure consists of a cathode (Zone 1) and a 3-dimensional conductive framework with gel polymer (Zone 2). According to them, the CNT/gel polymer layer inhibits the shuttle effect of conversion-type active materials. As an active material, the authors demonstrated the system using NiTe nanorods. Despite the fact that the electrochemical performance appeared to be enhanced, the paper is deemed to have little significance.

1) The 3-D conductive framework layer on the conversion-type cathode for adsorption-redox has been reported in numerous Li-S battery-related publications.

- Energy Environ. Sci., 2014, 7, 3381-3390 (Super P)
- J. Phys. Chem. Lett., 2014, 5, 1978-1983 (MWCNT)
- Small, 2016, 12, 174-179 (SWCNT)
- Carbon, 2016, 99, 624-632 (GO/CNT)
- ACS Appl. Mater. Interfaces, 2017, 9, 38445-38454 (hydroxyl SWCNT)
- Carbon, 2017, 122, 635-642 (N-doped MWCNT)

In addition, the 3-D conductive framework was applied to AIBs in other paper (Energy Environ. Sci., 2019, 12, 1918-1927). The only distinction is that the gel polymer was additionally used to further suppress the shuttle effect, but the novelty is weak when considering the impact of Nature Communications.

2) Charge balance in the conversion reaction of NiTe cathode requires AlCl_4^- anions, not Al^{3+} cations. This reaction is entirely distinct from the Li-S system, which employs Li^+ ions for a rocking-chair reaction independent of the amount of electrolyte. Although NiTe cathodes demonstrated a relatively high capacity of 1200 mAh/g, the actual energy density must include the amount of electrolyte since the system is not of the rocking-chair reaction. Refer to the publications for energy density calculation (Chem. Mater., 2017, 29, 4484-4492, and Nat. Energy, 4, 51-59). This type of conversion reaction cannot provide a competitive energy density, in my opinion.

Reviewer #2 (Remarks to the Author):

Suppressing the shuttle effect of the positive electrode materials in the strongly acidic chloroaluminate electrolytes is vital in improving the electrochemical performance of rechargeable AIBs. The manuscript by Yu et al. presented cathode-electrolyte engineering for constructing a layered quasi-solid AIB (LQS-AIB) and suggested a strategic designing of double micro quasi-solid reaction zones (DRZs) for improving the performances of AIB. Experimentally, the authors showed that such DRZs help to accelerate the redox reaction kinetics and reduce the shuttle effect of the dissolved active materials, resulting in a long-term cycling for AIBs. This is a nice study on the design of electrode-electrolyte for improving AIBs. The topic of this work would be interesting to the battery community in general. Therefore, I recommend acceptance of the manuscript for publication in Nature Communications after following issues are addressed.

1. As a key part of DRZs, the functionalized SWCNTs have significant effects on suppressing the shuttle effect of NiTe positive electrode. Although DFT calculations have demonstrated the adsorption capacity of functionalized SWCNTs to soluble ions, more detailed information of SWCNTs should be provided to further elucidate their roles, such as the size of SWCNTs and the contents of functional groups in the SWCNTs.

2. The CV curves in Figure 2A display four obvious couples of redox peaks. What reaction process does each redox peak correspond to? Why do the CV curves of LQS-AIB show an extra pair of redox peaks than those of QS-AIB (Supplementary Fig. 13)? Please clarify the detailed conversion processes and the related reaction mechanism based on the DRZs.

3. The analysis of EIS results in Figure 3i needs to be improved. The fitting curves and chi-squared values should be provided to help evaluate the fitting quality. The applied potential and amplitude should also be provided in Experimental Method.

4. Regarding the roles of DRZs, the authors have conducted ex-situ characterizations to verify the effect of DRZs on suppressing shuttle effect. However, the in-situ experiments should be more powerful to support the main arguments of this manuscript. If some in-situ characterizations can be provided, the quality of this work will be further improved.

5. The authors employ DRT (e.g. Figure 3j) to distribute the electrochemical process in time domain view. Please provide more detailed information about DRT transition, including the tools/software used for DRT transition.

6. Please provide scale bar for CT images in Figure 3 and optical images in Figure 4.

7. Although the NiTe material has been reported in a previous article, it is better to provide some characterization results on the structure and morphology of this material for the convenience of the readers.

Response to Reviewers' Comments

We would like to thank the reviewers for their valuable comments. We have studied the comments carefully and have made correction which we hope to meet with approval. The following is our point-to-point response to the concerns raised. For ease of reference, the reviewers' comments and suggestions are reproduced in **blue**, our response is in **black**, and the resulting changes to the manuscript are in **red**.

Reviewer #1 (Remarks to the Author):

This manuscript describes a layered quasi-solid structure for a high-capacity conversion-type cathode. The layered structure consists of a cathode (Zone 1) and a 3-dimensional conductive framework with gel polymer (Zone 2). According to them, the CNT/gel polymer layer inhibits the shuttle effect of conversion-type active materials. As an active material, the authors demonstrated the system using NiTe nanorods. Despite the fact that the electrochemical performance appeared to be enhanced, the paper is deemed to have little significance.

Reply: We feel great thanks for the reviewer's thorough reading and insightful feedback. The critical comments would definitely help us to further elaborate on the design concept of the double micro quasi-solid reaction zones (DRZs) and improve the quality of our manuscript. We have made a point-by-point response as shown below and hope that we have addressed the concerns of the reviewer.

Comment 1: The 3-D conductive framework layer on the conversion-type cathode for adsorption-redox has been reported in numerous Li-S battery-related publications.

- Energy Environ. Sci., 2014, 7, 3381-3390 (Super P)
- J. Phys. Chem. Lett., 2014, 5, 1978-1983 (MWCNT)
- Small, 2016, 12, 174-179 (SWCNT)
- Carbon, 2016, 99, 624-632 (GO/CNT)
- ACS Appl. Mater. Interfaces, 2017, 9, 38445-38454 (hydroxyl SWCNT)
- Carbon, 2017, 122, 635-642 (N-doped MWCNT)

In addition, the 3-D conductive framework was applied to AIBs in other paper (Energy Environ. Sci., 2019, 12, 1918-1927). The only distinction is that the gel polymer was additionally used to further

suppress the shuttle effect, but the novelty is weak when considering the impact of Nature Communications.

Reply: We appreciate the reviewer's critical comment. We also thank the reviewer for mentioning the previous papers on the 3D conductive framework for adsorption-redox and have carefully studied these works. However, we would like to interpret the differences between our work and the mentioned papers. Generally, the conversion-type positive electrode materials are easy to dissolve and shuttle in the chloroaluminate ionic liquid (IL) electrolytes with strong Lewis acid, resulting in rapid capacity decay. A promising approach is to encapsulate the chloroaluminate within a quasi-solid system to reduce the dissolution and diffusion rate of active materials. However, through previous research, we found that the electrochemical performance of conversion-type positive electrode materials in AIBs did not show significant improvement for two reasons: (1) The poor wetting ability of the quasi-solid electrolyte to active materials leads to an unfavorable electrode/electrolyte interface; (2) Although both dissolution and diffusion rates are decreased, the dissolved active materials would still undergo shuttle effects. In other words, the utilization rate of active materials is low in a conventional QS-AIB. **(Design target)**

To effectively address these issues, the main innovation of our work is to develop a quasi-solid AIB with double micro quasi-solid reaction zones (DRZs). The gel precursor solution penetrates into the positive electrode and polymerizes to form the first micro quasi-solid reaction zone (Zone 1), effectively reinforcing the wetting ability of the quasi-solid electrolyte to the active materials, thus greatly accelerating the redox reaction kinetics. In the second micro quasi-solid reaction zone (Zone 2), the single-walled carbon nanotubes (SWCNTs) with oxygen-containing functional groups are introduced into the polyacrylamide (PAM)-based gel (C-gel). Noticeably, the SWCNTs serve as both the adsorption additive and electronic conductor to establish a robust 3D conductive framework with the PAM-based gel in Zone 2, for fixing the active materials dissolved from Zone 1 and simultaneously providing sufficient ion and electron transport paths for further reaction. Therefore, Zone 2 shows a composite 3D conductive structure formed by solidifying a gel network for ion conduction and a carbon nanotube network as electronic conductor. This particular structure with double micro reaction zones can not only suppress the shuttle effect of the dissolved active materials, but also realize the maximum electron transfer of the positive electrode materials by accelerating reaction kinetics for the active materials in Zone 1 and continuing electrochemical reaction with the active materials dissolved to Zone 2. At the same time, the homogeneous gel that runs through Zone 1, Zone 2 and quasi-solid electrolyte constructs efficient ion channels, improving the electrode/electrolyte interface. **(Composition and working mechanism)**

In the aforementioned papers (Energy Environ. Sci., 2014, 7, 3381-3390; J. Phys. Chem. Lett., 2014, 5, 1978-1983; Small, 2016, 12, 174-179; Carbon, 2016, 99, 624-632; ACS Appl. Mater. Interfaces, 2017, 9, 38445-38454; Energy Environ. Sci., 2019, 12, 1918-1927), the authors reported a series of 3D conductive frameworks formed on the surface of separators using different carbon materials, referred to modified separators, to address the shuttle effect of active materials in liquid electrolytes. Although these modified separators suppressed the shuttle effect to some extent, some intrinsic problems still existed that cannot be solved in this way, especially in AIBs systems with strong acidic IL electrolytes. The explanations are as follows.

Due to the rapid dissolution of conversion-type active materials in the strong acidic IL electrolytes in AIBs, a large amount of active materials would dissolve into the electrolyte from positive electrode and accumulate on the surface of modified separators. After a certain amount of cycles, the contact interface between positive electrode and modified separators (3D conductive framework) becomes poor, even forming an electron-insulated space (filled with IL electrolytes) between them. Therefore, the electronic conductivity would significantly decrease between the accumulated active materials on the modified separators and the current collectors (or to say the positive electrode). In this case, these accumulated active materials on the modified separators turn to inactive materials, resulting in a significant capacity decay. While in our work, the homogeneous gel that runs through Zone 1, Zone 2 and quasi-solid electrolyte, and the conductive agent runs through Zone 1 and Zone 2 all along, guaranteeing stable ionic and electronic transport paths during long-term cycling.

To clearly demonstrate the differences between the DRZs in our work and the modified separators, we have also prepared a modified separator using the same acidified SWCNTs applied in Zone 2 (Supplementary Fig. 20). The electrochemical performance of AIBs using NiTe positive electrode and the modified separator is shown in Supplementary Fig. 21, which is much inferior to that based on DRZs.

(Battery performance)

Supplementary Fig. 20. SEM images of the modified separator based on acidified SWCNTs.

Supplementary Fig. 21. Cycling performance of AIB with modified separator (MS-AIB).

For the work reported in Carbon, 2017, 122, 635-642, the authors developed a 3D clew-like multiwalled carbon nanotube, which is actually a sulfur-loading host that serves as a conductive matrix. This is a very common approach for optimizing the sulfur positive electrodes in Li-S batteries, which is unrelated to the strategy of constructing modified separators or the DRZs in our work.

In brief, although we all require carbon materials in the preparation process, the design target, composition, working mechanism and battery performance of the DRZs constructed in our work are all different from the above-mentioned 3D conductive frameworks in the previous literatures. And we have emphasized these novelties to give weight to this paper, and made modifications to Scheme 1. The updated contents have been highlighted in the revised manuscript and Supplementary Fig. 20-21. Please see the following:

(Page 1) *“To address such issues, herein a layered quasi-solid AIB (LQS-AIB) system is developed with double micro reaction zones (DRZs, Zone 1 and Zone 2) to suppress dissolution and shuttle effect of active materials. Zone 1 is designed to accelerate the reaction kinetics by improving the wetting ability of quasi-solid electrolyte to active materials. Zone 2, constructed as a composite three-dimensional (3D) conductive framework interwoven by gel network for ion conduction and carbon nanotube network as electronic conductor, can fix the active materials dissolved from Zone 1 to allow for continuing electrochemical reactions.”*

(Page 3) *“Therefore, a promising approach is to encapsulate the chloroaluminate ILs within a quasi-solid system to reduce the dissolution and diffusion rate of active materials. Nevertheless, it is still hard to employ the quasi-solid AIB (QS-AIB) systems to effectively enhance the cycling performance of conversion-type positive electrode materials due to the unfavorable electrode/electrolyte interface and the existing partial dissolution of active materials (Scheme 1b). Therefore, no obvious battery performance improvement is observed for the low utilization of active materials in QS-AIB.”*

(Page 4) *“This particular structure with double micro reaction zones can not only suppress the shuttle effect of the dissolved active materials, but also realize the maximum electron transfer of the positive electrode materials by accelerating reaction kinetics of the active materials in Zone 1 and continuing electrochemical reaction with the active materials dissolved to Zone 2.”*

(Page 9) *“Additionally, the initial capacity of QS-AIB is also much lower than that of LQS-AIB. This is primarily attributed to the inadequate wetting ability of quasi-solid electrolyte to NiTe positive electrode and partial dissolution of the original active materials. These factors hinder the complete involvement of the active materials in the electron transfer process during both charge and discharge stages, ultimately leading to a low utilization rate of active materials. In contrast, these problems are addressed in LQS-AIB by constructing the DRZs, improving both the capacity and stability of the battery.”*

(Page 9, 10) *“In order to further compare the enhancing effect of the DRZs on battery performance, the cycling stability was also tested for AIB with modified separator (MS-AIB) based on the same SWCNTs (Supplementary Fig. 20). It is obvious that the electrochemical performance of MS-AIB is much inferior*

to that based on DRZs (Supplementary Fig. 21). This further highlights the effectiveness of constructing the DRZs to form the LQS-AIB as an optimization strategy.”

Comment 2: Charge balance in the conversion reaction of NiTe cathode requires AlCl_4^- anions, not Al^{3+} cations. This reaction is entirely distinct from the Li-S system, which employs Li^+ ions for a rocking-chair reaction independent of the amount of electrolyte. Although NiTe cathodes demonstrated a relatively high capacity of 1200 mAh/g, the actual energy density must include the amount of electrolyte since the system is not of the rocking-chair reaction. Refer to the publications for energy density calculation (Chem. Mater., 2017, 29, 4484-4492, and Nat. Energy, 4, 51-59). This type of conversion reaction cannot provide a competitive energy density, in my opinion.

Reply: Thanks so much for the reviewer’s insightful comment. We agree with the reviewer that the charge balance of redox reactions in AIBs is different from that of rocking-chair batteries, more similar to that of dual-ion batteries (*Nature* **520**, 324–328, doi: 10.1038/nature14340 (2015); *Adv. Energy Mater.* **9**, 1901749, doi: 10.1002/aenm.201901749 (2019); *Nat. Commun.* **12**, 3106, doi: 10.1038/s41467-021-23369-5 (2021); *Joule* **2**, 2528–2550, doi: 10.1016/j.joule.2018.09.003 (2018); *Angew. Chem. Int. Ed.* **59**, 3802–3832, doi: 10.1002/anie.201814294 (2020)), and the calculated energy density for this type of battery is relatively lower when the mass of electrolyte is considered.

In this work, we employed the conversion-type positive electrode and quasi-solid electrolyte, which enhanced the energy density of the AIB system from two aspects: (1) The conversion process with multi-electron transfer enhances the specific capacity of the positive electrode by approximately ten times compared to graphite materials and reduces the consumption of AlCl_3 required for a given charge storage; (2) The proportion of AlCl_3 in the electrolyte is increased and it is wrapped in the polymer framework by forming a quasi-solid electrolyte, which greatly reduces the risk of corrosion and moisture sensitivity from the high-proportion AlCl_3 . We have calculated the energy density of the conversion-type Al-NiTe battery in the revised supplementary information based on the method suggested by the reviewer. The calculation procedure is as follows:

It is known from the publications (Chem. Mater., 2017, 29, 4484-4492, and Nat. Energy, 2019, 4, 51-59) that the total capacity taking into account the mass of electrolyte in ionic liquid system ($\text{AlCl}_3/[\text{EMIm}]\text{Cl}$) can be calculated based on the following formula:

$$C_{\text{total}} = \frac{C_{\text{ne}} C_{\text{po}}}{C_{\text{ne}} + C_{\text{po}}} = \frac{Fx(r-1)C_{\text{po}}}{Fx(r-1) + C_{\text{po}}(rM_{\text{AlCl}_3} + M_{[\text{EMIm}]\text{Cl}})} \quad (1)$$

where F is Faraday constant (26.8×10^3 mAh mol⁻¹), x is number of electrons used to reduce 1 mol of AlCl₃, r is the AlCl₃/[EMIm]Cl molar ratio, C_{ne} and C_{po} are respectively the specific capacity of the actual negative electrode material (AlCl₃) and positive electrode material (mAh g⁻¹), and M is the molar mass (g mol⁻¹).

As for the quasi-solid electrolyte system, the molar ratio of AlCl₃: [EMIm]Cl: acrylamide is 2:1:0.4. Therefore, the specific capacity of the actual negative electrode material AlCl₃ in the gel polymer electrolyte can be expressed as:

$$C_{ne} = \frac{Fx(r_{AlCl_3} - r_{[EMIm]Cl} - r_{acrylamide})}{r_{AlCl_3}M_{AlCl_3} + r_{[EMIm]Cl}M_{[EMIm]Cl} + r_{acrylamide}M_{acrylamide}} \quad (2)$$

In addition, batteries that use NiTe as a positive electrode material operate based on the following reaction:

According to the above reaction, the value of x for NiTe is 2, which means that the NiTe positive electrode requires less electrolyte for the given charge storage. And the total capacity and energy density of a conversion-type Al-NiTe battery are as follows:

$$C_{total} = 69 \text{ mAh g}^{-1}; \text{ Energy density} = 90 \text{ Wh kg}^{-1} \text{ (average voltage} = 1.3 \text{ V)}$$

For comparison, the energy density of an Al-graphite battery in the same quasi-solid system is also calculated to be 44 Wh kg⁻¹ (based on $C_{graphite} = 120$ mAh g⁻¹, average voltage = 2 V). Meanwhile, it can be concluded from the publication (Chem. Mater., 2017, 29, 4484-4492) that in the ionic liquid electrolyte system, AlCl₃/[EMIm]Cl = 1.3 is considered the optimal molar ratio for ILs-AIB in most reported papers (Prog. Mater. Sci. 128, doi: 100960, 10.1016/j.pmatsci.2022.100960 (2022)), in which the highest energy density of the Al-graphite battery is 33 Wh kg⁻¹, much lower than that of the Al-NiTe battery with double micro reaction zones after 4000 cycles in this work (80 Wh kg⁻¹). In addition, we can clarify from the publication (Nat. Energy, 2019, 4, 51-59) that the competitive energy density of organic positive electrode material is ~54 Wh kg⁻¹, which is also lower than that of AIBs based on the NiTe conversion-type positive electrode in our work. Therefore, we believe that the conversion-type positive electrode materials based on multi-electron transfer, with the employment of quasi-solid system, have a significant potential for energy density improvement in AIBs.

It is worth noting that if the energy density is calculated based on the total mass of the battery device, the separator mass as an additional component is needed to be included in ILs-AIBs. In the quasi-solid system, acrylamide plays a dual role. It undergoes complex dissociation with AlCl₃ and subsequently

serves as the electrolyte framework after polymerization. It is important to note that we have already considered its mass in the calculation mentioned above. Meanwhile, the successful construction of Zone 1 enhances the wetting ability of the electrolyte to the active material and the quasi-solid gel electrolyte exhibits high flexibility and controllable thickness. Therefore, there is no need to consider additional electrolyte mass for wetting electrode materials and separators, as required in liquid rocking-chair batteries or ILs-AIBs. Additionally, unlike rocking-chair batteries, the AIBs are a negative-electrode-free system. In practical application, a minimally thin Al film can guarantee the batteries normal operation. Thus, we believe that the conversion-type AIBs can indeed provide a competitive energy density with the assistance of the quasi-solid electrolyte system.

More importantly, the development of AIBs is primarily focused on large-scale high-safety energy storage applications, where the requirements for energy density are not as urgent as the requirements for safety and cost-effectiveness. In fact, due to the abundant reserve of Al in the earth's crust and the nonflammable characteristics of electrolytes, AIBs have the potential to be a low-cost and high-safety energy storage device. Therefore, the development and enhancement of AIBs hold great significance in future.

The updated contents have been highlighted in the revised manuscript and supplementary note. Please see the following:

(Page 15) *“Most notably, it can be concluded that, even when considering the mass of the electrolyte, the conversion reactions with multi-electron transfer of conversion-type positive electrode such as NiTe are more competitive in energy density due to the less AlCl₃ required for the given charge storage (See details in Supplementary Note), compared with typical graphite or organic positive electrodes^{47,48}. Accordingly, the design of DRZs to construct the LQS-AIBs could be considered as an effective way to address the existing challenge of balancing high energy density and long cycle life in aluminum-ion batteries.”*

(Revised Supporting Information)

“Supplementary Note

Energy Density Calculation

The charge balance of redox reactions in AIBs is different from that of rocking-chair batteries, more similar to that of dual-ion batteries¹⁻⁵. It is known that the total capacity taking into account the mass of

the electrolyte in ionic liquid system (AlCl₃/[EMIm]Cl) can be calculated based on the following formula^{2,6-8}:

$$C_{\text{total}} = \frac{C_{\text{ne}} C_{\text{po}}}{C_{\text{ne}} + C_{\text{po}}} = \frac{Fx(r-1)C_{\text{po}}}{Fx(r-1) + C_{\text{po}}(rM_{\text{AlCl}_3} + M_{[\text{EMIm}]\text{Cl}})} \quad (1)$$

where F is Faraday constant ($26.8 \times 10^3 \text{ mAh mol}^{-1}$), x is number of electrons used to reduce 1 mol of AlCl₃, r is the AlCl₃/[EMIm]Cl molar ratio, C_{ne} and C_{po} are respectively the specific capacity of the actual negative electrode material (AlCl₃) and positive electrode material (mAh g⁻¹), and M is the molar mass (g mol⁻¹).

As for the quasi-solid electrolyte system, the molar ratio of AlCl₃: [EMIm]Cl: acrylamide is 2:1:0.4. Therefore, the specific capacity of the actual negative electrode material AlCl₃ in the gel polymer electrolyte can be expressed as:

$$C_{\text{ne}} = \frac{Fx(r_{\text{AlCl}_3} - r_{[\text{EMIm}]\text{Cl}} - r_{\text{acrylamide}})}{r_{\text{AlCl}_3} M_{\text{AlCl}_3} + r_{[\text{EMIm}]\text{Cl}} M_{[\text{EMIm}]\text{Cl}} + r_{\text{acrylamide}} M_{\text{acrylamide}}} \quad (2)$$

In addition, batteries that use NiTe as a positive electrode material operate based on the following reaction:

According to the above reaction, the value of x for NiTe is 2, which means that the NiTe positive electrode requires less electrolyte for the given charge storage. And the total capacity and energy density of a conversion-type Al-NiTe battery are as follows:

$C_{\text{total}} = 69 \text{ mAh g}^{-1}$; Energy density = 90 Wh kg^{-1} (average voltage = 1.3 V)

For comparison, the energy density of an Al-graphite battery in the same quasi-solid system is also calculated to be 44 Wh kg^{-1} (based on $C_{\text{graphite}} = 120 \text{ mAh g}^{-1}$, voltage = 2 V). Meanwhile, it can be concluded from the literature that in the ionic liquid electrolyte system⁶, AlCl₃/[EMIm]Cl = 1.3 is considered the optimal molar ratio for ILs-AIB in most reported papers⁹, in which the highest energy density of the Al-graphite battery is 33 Wh kg^{-1} , much lower than that of the Al-NiTe battery with double micro reaction zones after 4000 cycles in this work (80 Wh kg^{-1}). In addition, we can clarify from the literature⁷ that the energy density of organic positive electrode material is $\sim 54 \text{ Wh kg}^{-1}$, which is also lower than that of AIBs based on the NiTe conversion-type positive electrode in our work. Therefore, it can be concluded that the quasi-solid conversion-type AIBs are more competitive in energy density due to the less AlCl₃ required for the given charge storage and the higher proportion of AlCl₃ in the electrolyte. With the construction

of DRZs, the LQS-AIBs can continuously and steadily deliver an impressive energy density, demonstrating broad application prospects in the field of large-scale high-safety energy storage.”

The corresponding references:

47. Kim, D. J., et al. Rechargeable aluminum organic batteries. *Nat. Energy* **4**, 51-59 (2019).
48. Kravchyk, K. V., Wang, S., Piveteau, L. & Kovalenko, M. V. Efficient aluminum chloride-natural graphite battery. *Chem. Mater.* **29**, 4484–4492 (2017).

The corresponding supplementary references:

- S1. Lin, M.-C. et al. An ultrafast rechargeable aluminium-ion battery. *Nature* **520**, 324-328 (2015).
- S2. Kravchyk, K. V. & Kovalenko, M. V. Rechargeable dual-ion batteries with graphite as a cathode: Key challenges and opportunities. *Adv. Energy Mater.* **9**, 1901749 (2019).
- S3. Huang, Z. et al. Manipulating anion intercalation enables a high-voltage aqueous dual ion battery. *Nat. Commun.* **12**, 3106 (2021).
- S4. Placke, T. et al. Perspective on performance, cost, and technical challenges for practical dual-ion batteries. *Joule* **2**, 2528-2550 (2018).
- S5. Zhou, X. et al. Strategies towards Low-Cost Dual-Ion Batteries with High Performance. *Angew. Chem. Int. Ed.* **59**, 3802-3832 (2020).
- S6. Kravchyk, K. V., Wang, S., Piveteau, L. & Kovalenko, M. V. Efficient aluminum chloride-natural graphite battery. *Chem. Mater.* **29**, 4484–4492 (2017).
- S7. Kim, D. J. et al. Rechargeable aluminium organic batteries. *Nat. Energy* **4**, 51–59 (2019).
- S8. Kravchyk, K. V. & Kovalenko, M. V. Aluminum electrolytes for Al dual-ion batteries. *Commun. Chem.* **3**, 1-9 (2020).
- S9. Han, X. et al. Electrolytes for rechargeable aluminum batteries. *Prog. Mater. Sci.* **128**, 100960 (2022).

Reviewer #2 (Remarks to the Author):

Suppressing the shuttle effect of the positive electrode materials in the strongly acidic chloroaluminate electrolytes is vital in improving the electrochemical performance of rechargeable AIBs. The manuscript by Yu et al. presented cathode-electrolyte engineering for constructing a layered quasi-solid AIB (LQS-AIB) and suggested a strategic designing of double micro quasi-solid reaction zones (DRZs) for improving the performances of AIB. Experimentally, the authors showed that such DRZs help to accelerate the redox reaction kinetics and reduce the shuttle effect of the dissolved active materials, resulting in a long-term cycling for AIBs. This is a nice study on the design of electrode-electrolyte for improving AIBs. The topic of this work would be interesting to the battery community in general. Therefore, I recommend acceptance of the manuscript for publication in Nature Communications after following issues are addressed.

Reply: We would like to thank the reviewer so much for the careful review and positive comments, which will help to improve the quality of our manuscript. Based on the reviewer's insightful suggestions, the manuscript has been thoroughly revised and all changes have been highlighted in red in the revised manuscript and supplementary information.

Comment 1: As a key part of DRZs, the functionalized SWCNTs have significant effects on suppressing the shuttle effect of NiTe positive electrode. Although DFT calculations have demonstrated the adsorption capacity of functionalized SWCNTs to soluble ions, more detailed information of SWCNTs should be provided to further elucidate their roles, such as the size of SWCNTs and the contents of functional groups in the SWCNTs.

Reply: Thanks for the reviewer's valuable comment. We have added detailed information of SWCNTs before and after acidification with the characterizations of SEM, EDS, XPS, Raman and zeta potential. The characterization results are shown in Supplementary Fig. 7-9, which have been added in the revised manuscript and supplementary information. The acidified SWCNTs for constructing DRZs show elemental composition of C and O, with a length of tens of micrometers. The contents of functional groups in the acidified SWCNTs are listed in Supplementary Table 3. The updated parts have been highlighted in the revised manuscript. Please see the following:

Supplementary Fig. 7. a,b SEM images and c EDS spectra of the original SWCNTs. d,e SEM images and f EDS spectra of the acidified SWCNTs.

Supplementary Fig. 8. a Survey XPS spectrum. b,c C 1s XPS spectra before and after acidification. d O 1s XPS spectra of the acidified SWCNTs.

Supplementary Fig. 9. a Raman spectra and **b** zeta potential of the SWCNTs before and after acidification.

Supplementary Table 3. Curve fitting results of C 1s XPS spectra.

SWCNTs	sp ² / %	sp ³ / %	-C-OH / %	-COOH / %	π-π* / %
Original	50.92	29.85	3.97	6.06	9.20
Acidified	41.10	32.28	11.75	9.20	5.67

(Page 5, 6) “Thus, SWCNTs were chemically modified by acid treatment²⁹ to attach the oxygen-containing functional groups. From the SEM images (Supplementary Fig. 7), the acidified SWCNTs with the rough outer wall and a length of tens of micrometers have better dispersion. And the metal impurity elements in the original SWCNTs are removed, and the oxygen content increases obviously (Supplementary Fig. 7c, d). In the X-ray photoelectron spectroscopy (XPS) survey scan of the SWCNTs samples (Supplementary Fig. 8a), pronounced peaks corresponding to C 1s and O 1s transition are observed for the acidified SWCNTs. The results of C 1s XPS spectra (Supplementary Fig. 8b, c) reveal the presence of peaks for SWCNTs corresponding to sp²-hybridized carbon (284.8 eV), sp³-hybridized carbon (285.1 eV), hydroxyl carbon (286.6 eV), carboxyl carbon (288.9 eV) and π-π* transitions at higher binding energy range (290.9 eV) in different relative percentages of each contribution before and after acidification^{30,31} (Supplementary Table 3). And the carboxyl and hydroxyl functional groups on acidified SWCNTs are also confirmed through the deconvolution of XPS O 1s peak (Supplementary Fig. 8d). The enhanced I_D/I_G ratio in Raman spectra (Supplementary Fig. 9a) also indicates the successful functionalization of the acidified SWCNTs³². Furthermore, the acidified SWCNTs reveal a negative zeta potential of -15 mV

(Supplementary Fig. 9b), which is highly beneficial to the adsorption of cations (such as Ni^{2+} , Te_n^{2+} and TeCl_3^+) dissolved in electrolyte.”

The corresponding references:

29. Chen, Y., et al. Insights into the biomedical effects of carboxylated single-wall carbon nanotubes on telomerase and telomeres. *Nat. Commun.* **3**, 1-13 (2012).

30. Ago, H. et al. Work functions and surface functional groups of multiwall carbon nanotubes. *J. Phys. Chem. B* **103**, 8116–8121 (1999).

31. Datsyuk, V., et al. Chemical oxidation of multiwalled carbon nanotubes. *Carbon* **46**, 833-840 (2008).

32. Liu, M., Yang, Y., Zhu, T. & Liu, Z. Chemical modification of single-walled carbon nanotubes with peroxytrifluoroacetic acid. *Carbon* **43**, 1470–1478 (2005).

Comment 2: The CV curves in Figure 2A display four obvious couples of redox peaks. What reaction process does each redox peak correspond to? Why do the CV curves of LQS-AIB show an extra pair of redox peaks than those of QS-AIB (Supplementary Fig. 13)? Please clarify the detailed conversion processes and the related reaction mechanism based on the DRZs.

Reply: Thanks so much for the reviewer’s helpful comment. Generally, the NiTe positive electrodes have four couples of redox peaks in both LQS- and QS-AIB, which correspond to the complex conversion processes of $\text{Te}^{2-} \leftrightarrow \text{Te} \leftrightarrow \text{Te}_n^{2+} (1 \leq n \leq 8) \leftrightarrow \text{TeCl}_3^+$. Herein, Te_n^{2+} exists in the form of polycationic clusters, such as Te_4^{2+} , Te_6^{2+} and Te_8^{2+} , which are stabilized by the large anions in Lewis acidic melts (*Inorg. Chem.* **11**, 2648-2652, doi: 10.1021/ic50117a018 (1972); *Angew. Chem. Int. Ed.* **33**, 163–172, doi: 10.1002/anie.199401631 (1994); *Angew. Chem. Int. Ed.* **55**, 1173–1177, doi: 10.1002/anie.201507644 (2016)). When the battery was charged to 1.5 V, the oxidation product was confirmed to be $\text{Te}_4(\text{AlCl}_4)_2$ in our previous work (*ACS Nano* **14**, 3469-3476 doi: 10.1021/acsnano.9b09550 (2020)). Therefore, it can be inferred that during the charging process, Te^{2-} is first converted to Te, and further oxidized to form long-chain $\text{Te}_n^{2+} (4 < n \leq 8)$, which is then converted to short-chain $\text{Te}_n^{2+} (1 < n \leq 4)$ to generate Te^{2+} , finally being oxidized to TeCl_3^+ under high potential. In the subsequent discharge process, the corresponding reversible reduction reactions occur. However, in QS-AIB, the reaction products of positive electrode are dissolved into the strong acidic electrolyte and permanently isolated from the current collector, making it unable to continue participating in the conversion reaction. This is manifested in the CV curves

(Supplementary Fig. 16) as the loss of the reduction peak A' and oxidation peak D, and thus only three redox pairs appear. In LQS-AIB, the dissolved reaction products can be fixed in Zone 2 with sufficient ion and electron transport channels to continue the reversible electrochemical reaction, thus displaying a complete four-pair of redox peaks in the CV curves. The description has been added in the revised manuscript. Please see the following:

Supplementary Fig. 16. CV curves of QS-AIB at a scan rate of 1 mV s^{-1} .

(Page 8) *“Four redox pairs are located at 2.13/1.84 V, 1.69/1.05 V, 1.23/0.62 V and 0.69/0.28 V for LQS-AIB, respectively, corresponding to the complex conversion processes of $\text{Te}^{2-} \leftrightarrow \text{Te} \leftrightarrow \text{Te}_n^{2+}$ ($1 \leq n \leq 8$) $\leftrightarrow \text{TeCl}_3^+$. Herein, Te_n^{2+} exists in the form of polycationic clusters, such as Te_4^{2+} , Te_6^{2+} and Te_8^{2+} , which are stabilized by the large anions in Lewis acidic melts³⁶⁻³⁸. When the battery was charged to 1.5 V, the oxidation product was confirmed to be $\text{Te}_4(\text{AlCl}_4)_2$ in our previous work. Therefore, it can be inferred that during the charging process, Te^{2-} is first converted to Te , and further oxidized to form long-chain Te_n^{2+} ($4 < n \leq 8$), which is then converted to short-chain Te_n^{2+} ($1 < n \leq 4$) to generate Te^{2+} , finally being oxidized to TeCl_3^+ under high potential. In the subsequent discharge process, the corresponding reversible reduction reactions occur. In the case of QS-AIB, the CV curves reveal the absence of reduction peak A' and oxidation peak D (Supplementary Fig. 16). It could be concluded that the reaction products formed at high or low potentials are more prone to be dissolved in the acidic electrolyte and get permanently isolated from the current collector. As a result, these dissolved products are unable to continue participating in the electrochemical reaction. Additionally, all peak currents gradually decrease with cycling, further confirming the dissolution of the active material in QS-AIB.”*

The corresponding references:

36. Bjerrum, N. J. Lower oxidation states of tellurium. III. Ditellurium (2+) in chloroaluminate melts. *Inorg. Chem.* **11**, 2648-2652 (1972).

37. Schulz, C., Daniels, J., Bredow, T. & Beck, J. The electrochemical synthesis of polycationic clusters. *Angew. Chem. Int. Ed.* **55**, 1173-1177 (2016).

38. Beck, J. New forms and functions of tellurium: From polycations to metal halide tellurides. *Angew. Chem. Int. Ed.* **33**, 163-172 (1994).

Comment 3: The analysis of EIS results in Figure 3i needs to be improved. The fitting curves and chi-squared values should be provided to help evaluate the fitting quality. The applied potential and amplitude should also be provided in Experimental Method.

Reply: Thanks so much for the reviewer's helpful comment. We have fitted the EIS data in Fig. 3i and updated the fitting curves and parameters in Supplementary Fig. 24 and Supplementary Table 4. Chi-squared values of both fitting QS- and LQS-AIB are lower than 10^{-3} , representing good fitting quality. Additionally, we have also supplemented the applied potential and amplitude in the Method part. The updated contents are highlighted in the revised manuscript. Please see the following:

Supplementary Fig. 24. Nyquist plots of the original QS-, LQS-AIBs and the corresponding fitting curves.

Supplementary Table 4. Parameters obtained by fitting the impedance spectra of the original QS- and LQS-AIBs.

	QS-AIB	LQS-AIB
R_s / Ω	5.54	11.3
R_{ct1} / Ω	84.19	115.9
CPE_{1-T} / F	2.44E-5	1.72E-5
CPE_{1-P} / F	0.86	0.67
R_{ct2} / Ω	724.3	519.2
CPE_{2-T} / F	1.27E-4	2.31E-4
CPE_{2-P} / F	0.77	0.86
Chi-squared values	0.00097	0.00076

(Page 11) *“The applicability of the equivalent circuit model has been verified by Chi-square values, which are lower than 10^{-3} (Supplementary Table 4), representing good fitting quality.”*

(Page 18) *“Electrochemical impedance spectroscopy (EIS) was carried out with a CHI 660E electrochemical workstation (Shanghai, China) at the frequency range of 100 kHz ~ 0.01 Hz with oscillation amplitude of 5 mV under the open circuit potential.”*

Comment 4: Regarding the roles of DRZs, the authors have conducted ex-situ characterizations to verify the effect of DRZs on suppressing shuttle effect. However, the in-situ experiments should be more powerful to support the main arguments of this manuscript. If some in-situ characterizations can be provided, the quality of this work will be further improved.

Reply: Thanks for the reviewer’s nice suggestion. Since the whole battery system, especially the DRZs, is wrapped by gel polymer electrolytes, it is very difficult to obtain material information through in-situ characterizations. However, we can verify the role of DRZs on suppressing shuttle effect for visualization by conducting permeation experiments. As shown in Supplementary Fig. 29, H-type cells were employed, in which the IL electrolyte with dissolved active materials (left chamber) and the blank IL electrolyte (right chamber) were separated by GF/A separator, GPE and C-gel/GPE, respectively, to simulate the

shuttle process within the ILs-AIB, QS-AIB and LQS-AIB. The results after different settling time can be very intuitive, indicating the effective suppression of the shuttle effect with the construction of DRZs. The updated experiments and descriptions have been added in the revised manuscript and supplementary information. Please see the following:

Supplementary Fig. 29. Permeation tests for the three types of AIBs.

(Page 13, 14) *“To further verify the role of DRZs on suppressing shuttle effect, permeation experiments were conducted using the H-type cells for visualization (Supplementary Fig. 29), in which the IL electrolyte with dissolved NiTe active materials (left chamber) and the blank IL electrolyte (right chamber) were separated by GF/A separator, GPE and C-gel/GPE, respectively, to simulate the diffusion process within the ILs-AIB, QS-AIB and LQS-AIB. The results after settling for a period of time indicate that the purple dissolved materials can easily pass through the GF/A separator within 1 minute. Consequently, the blank electrolyte in the right chamber darkens in color after only 12 h. And the diffusion rate of purple dissolved materials is slowed down when employing GPE. For the case of C-gel/GPE,*

almost no visible change in color of blank IL electrolyte is observed even after 36 h, indicating excellent performance of LQS-AIB in suppressing shuttle effects.”

Comment 5: The authors employ DRT (e. g. Figure 3j) to distribute the electrochemical process in time domain view. Please provide more detailed information about DRT transition, including the tools/software used for DRT transition.

Reply: Thanks for the reviewer’s helpful suggestions. The RelaxIS software was applied for DRT transition from the EIS data, which has been clarified in the Method part in the revised manuscript. Please see the following:

(Page 18) *“And the transition for distribution of relaxation times (DRT) was conducted using RelaxIS impedance spectrum analysis.”*

Comment 6: Please provide scale bar for CT images in Figure 3 and optical images in Figure 4.

Reply: We would like to thank the reviewer for the helpful suggestions. The scale bar for CT images in Fig. 3 and optical images in Fig. 4 has been added in the revised manuscript. Please see the following:

Fig. 3 | Stable interface in the layered structure of LQS-AIB. *Ex-situ* X-ray nano-computed tomography (CT) images of **a** reconstructed architecture of Zone 1 in QS-AIB and **b** front, **c** left and **d** top view of the corresponding profiles. **e** Reconstructed architecture of the DRZs in LQS-AIB and **f** front, **g** left and **h** top view of the corresponding profiles of DRZs. **i** Comparison of Nyquist plots for the LQS- and QS-AIBs. Inset shows the equivalent circuit model for the battery systems. **j** DRT results of LQS- and QS-AIBs. **k** Comparison of the performance of our LQS-AIB with the other Al batteries based on conversion-type positive electrode materials.

Fig. 4 | Suppression of shuttle effect. **a** CV curves of the LQS- and QS-AIB after 200 cycles. **b** Charge and discharge curves for the two battery systems at the 20th and 200th cycles. **c** Schematic diagram of the different points in the cycled QS-AIB and **d** corresponding Te 3d XPS spectra and **e** EDS spectra at point 2. **f** Schematic diagram of the different points in the cycled LQS-AIB and **g** corresponding Te 3d XPS spectra and **h** EDS spectra at point 2. **i** Optical images of Al negative electrodes before and after cycles in the ILs-, QS- and LQS-AIBs.

Comment 7: Although the NiTe material has been reported in a previous article, it is better to provide some characterization results on the structure and morphology of this material for the convenience of the readers.

Reply: Thanks a lot for the helpful comment. Based on reviewer's suggestion, we have added the characterizations on structure and morphology of NiTe positive material. The SEM, TEM images and XRD spectra, as well as the corresponding descriptions, have been added in the revised manuscript and supplementary information (Supplementary Fig. 2). Please see the following:

Supplementary Fig. 2. Characterizations of NiTe nanorods. **a** XRD spectra. **b,c** SEM images. **d-f** TEM images.

(Page 5) *“The X-ray diffraction (XRD) pattern of the positive electrode material could be assigned to the hexagonal NiTe phase (JCPDS No. 65-9466, Supplementary Fig. 2a) and the field emission scanning electron microscopy (FE-SEM) images suggest its uniform rod-like morphology (Supplementary Fig. 2b-c). Additionally, transmission electron microscopy (TEM) images demonstrate that the nanorods are composed of many black nanoclusters (Supplementary Fig. 2d-f).”*

Hopefully, our Reply and revision have well cleared all the concerns upon reading the manuscript. Again, many thanks for help in improving our work.

Sincerely yours,

Shuqiang Jiao

Professor of Electrochemical Engineering

State Key Laboratory of Advanced Metallurgy

University of Science and Technology Beijing, Beijing 100083, China.

Email: sjiao@ustb.edu.cn

Tel&Fax: +86-10-62333617

Jul. 2, 2023

REVIEWERS' COMMENTS

Reviewer #2 (Remarks to the Author):

Authors have addressed all my previous concerns and improved the manuscript quality. I therefore recommend the manuscript to be accepted at Nature communications in its current form.

Response to Reviewers' Comments

We would like to thank the reviewers for their valuable comments. We are delighted that the reviewers have recommend our manuscripts for publication in Nature Communications. For ease of reference, the reviewers' comments and suggestions are reproduced in **blue**, and our response is in **black**.

Reviewer #2 (Remarks to the Author):

Authors have addressed all my previous concerns and improved the manuscript quality. I therefore recommend the manuscript to be accepted at Nature communications in its current form.

Reply: We sincerely thank the reviewer for the valuable evaluation and positive feedback.

Again, many thanks for help in improving our work.

Sincerely yours,

Shuqiang Jiao

Professor of Electrochemical Engineering

State Key Laboratory of Advanced Metallurgy

University of Science and Technology Beijing, Beijing 100083, China.

Email: sjiao@ustb.edu.cn

Tel&Fax: +86-10-62333617

Aug. 11, 2023